# Measuring instability in chronic human intracortical neural recordings towards stable, long-term brain-computer interfaces

Tsam Kiu Pun [1,2,3] ✉, Mona Khoshnevis[4], Tommy Hosman[2,5], Guy H. Wilson[6], Anastasia Kapitonava[7], Foram Kamdar[6], Jaimie M. Henderson [6,8], John D. Simeral[2,3,5], Carlos E. Vargas-Irwin [3,5,9], Matthew T. Harrison[3,4,11] & Leigh R. Hochberg [2,3,5,7,10,11]

Intracortical brain-computer interfaces (iBCIs) enable people with tetraplegia to gain intuitive cursor control from movement intentions. To translate to practical use, iBCIs should provide reliable performance for extended periods of time. However, performance begins to degrade as the relationship between kinematic intention and recorded neural activity shifts compared to when the decoder was initially trained. In addition to developing decoders to better handle long-term instability, identifying when to recalibrate will also optimize performance. We propose a method, "MINDFUL", to measure instabilities in neural data for useful long-term iBCI, without needing labels of user intentions. Longitudinal data were analyzed from two BrainGate2 participants with tetraplegia as they used fixed decoders to control a computer cursor spanning 142 days and 28 days, respectively. We demonstrate a measure of instability that correlates with changes in closed-loop cursor performance solely based on the recorded neural activity (Pearson $r = 0.93$ and $0.72$, respectively). This result suggests a strategy to infer online iBCI performance from neural data alone and to determine when recalibration should take place for practical long-term use.

Intracortical brain-computer interfaces (iBCIs) have enabled people with tetraplegia to control external devices by decoding movement intentions from neural recordings[1–7]. iBCIs can also restore communication by providing rapid point-and-click cursor control for applications such as typing, web browsing and navigating apps on a tablet[8–13], and can enable speech-to-text decoding for people with severe dysarthria[14]. Decoders are typically trained during explicit calibration epochs that allow for simultaneous collection of recorded neural signals during instructed motor intentions[6,15–17]. After training, decoding performance varies over time because of complex biological and device-related instabilities that are not fully understood[18,19]. Persistent periods of decreased performance are commonly observed with

existing decoding paradigms and remain one of the challenges that hinders wider adoption of iBCIs for people with paralysis[12,18–22]. Restoration of good control after performance has degraded often requires the user to repeat a calibration task to retrain the decoder[5,18]. Reducing the frequency and duration of explicit recalibration tasks are important for improving the utility of iBCIs. A step in this direction would be a method that can monitor performance and automatically determine when recalibration or other measures are necessary.

Here, we show that decoding performance on timescales of tens of seconds can be estimated from the same recorded neural signal used for motor decoding. The foundational principle is that persistent changes in

[1]Biomedical Engineering Graduate Program, School of Engineering, Brown University, Providence, RI, USA. [2]School of Engineering, Brown University, Providence, RI, USA. [3]Carney Institute for Brain Science, Brown University, Providence, RI, USA. [4]Division of Applied Mathematics, Brown University, Providence, RI, USA. [5]VA RR&D Center for Neurorestoration and Neurotechnology, Rehabilitation R&D Service, Providence VA Medical Center, Providence, RI, USA. [6]Department of Neurosurgery, Stanford University, Stanford, CA, USA. [7]Center for Neurotechnology and Neurorecovery, Department of Neurology, Massachusetts General Hospital, Boston, MA, USA. [8]Wu Tsai Neurosciences Institute and Bio-X Institute, Stanford University, Stanford, CA, USA. [9]Department of Neuroscience, Brown University, Providence, RI, USA. [10]Department of Neurology, Harvard Medical School, Boston, MA, USA. [11]These authors jointly supervised this work: Matthew T. Harrison, Leigh R. Hochberg. ✉e-mail: tsam_kiu_pun@brown.edu

performance likely result from statistical changes in the recorded neural signals. Measures of such statistical changes might then be a good surrogate for measures of performance changes. We call this approach "**MINDFUL**" (**m**easuring **in**stabilities in **n**eural **d**ata **f**or **u**seful **l**ong-term iBCI). More specifically, given a target period for which average decoding performance is unknown, we calculate a statistical distance between the distribution of neural activity patterns during the target period and a similar distribution collected when performance was known to be good (such as when the decoder was first trained), as illustrated in Fig. 1a. The MINDFUL score obtained using Kullback–Leibler divergence (KLD) to compare neural activity patterns was found to correlate with decoding performance.

Changes in decoding performance can be attributed to many types of variability in the recorded neural signals. BCI decoding algorithms typically model rapid fluctuations in neural features (on timescales of tens of milliseconds) that have no apparent correlation with motor intention as stochastic *noise*. Noise is a useful explanation for why decoding performance changes from instant to instant, but it generally does not account for persistent changes in (average) decoding performance that last seconds or more. We ascribe such persistent changes to *model drift*, which we define as changes in the relationship between recorded neural signals and motor intention. Nonstationarity, feature shift, and dataset shift are terms that have been used synonymously in the literature for this type of phenomenon, but they sometimes refer to any changes in the recorded signals rather than changes in the signal-decoder relationship[23–30]. Model drift can be attributed to various factors such as changes in action potential waveforms[18,31,32], neural tuning profiles[33–35], cognitive strategy or plasticity due to learning[36–38], material degradation and tissue responses to the recording device[39,40], and array micro-movements[41]. The type and magnitude of model drift result in various forms of performance degradation[19], sometimes necessitating decoder recalibration to restore control. Existing solutions to reduce the need for recalibration tasks include adaptive decoders that require shorter recalibration sessions to maintain or restore stable performance[3,10,42], self-supervised recalibration using retrospective labeling that avoids explicit recalibration sessions[26,27,43,44], and robust decoders that experience less model drift by extracting stable, time-invariant features from high-dimensional recordings[20–22,45–51] or by adaptively adjusting decoder parameters[12,52].

The model drift that influences performance is necessarily a property of the joint distribution of recorded neural signals and motor intention. It is not a priori clear, however, that model drift related to performance can be meaningfully identified from the recorded neural signals alone, which is what MINDFUL attempts to capture. MINDFUL differs from previously studied statistical tests for model drift that additionally require knowledge of movement intention[28,53,54]. Since true movement intention is often unavailable in iBCI applications where people with paralysis control an external device without being cued to acquire targets (e.g., a cursor on a tablet computer being used to send an email[22]), an approach like MINDFUL based

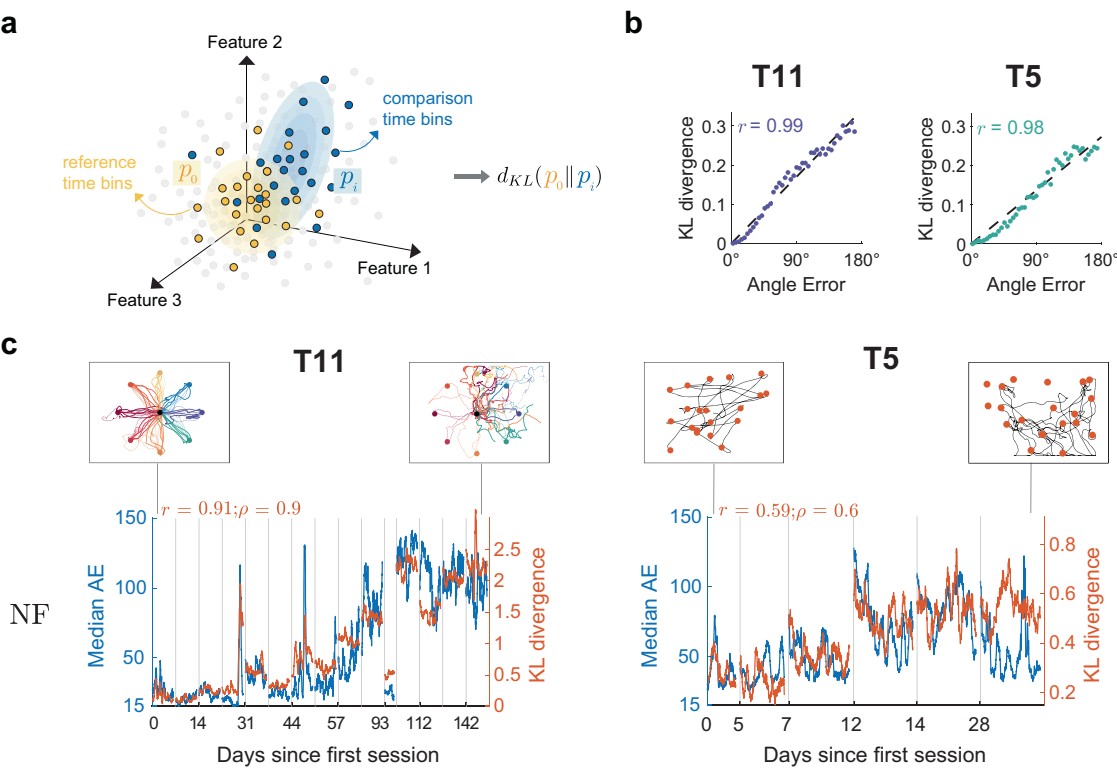

**Fig. 1 | MINDFUL score correlates with performance over time. a** Illustration of how the MINDFUL metric is calculated. Each dot symbolizes neural features at a given time bin in any given session, colored by whether the time bin is included for estimating the reference distribution (yellow), or for comparison (blue), or neither (gray). The difference in distributions is quantified by Kullback–Leibler divergence (KLD) between the reference distribution, $p_0$ and the comparison distribution, $p_i$. **b** Binned samples of neural features were grouped according to decoder performance in terms of angle error (AE), including data from all sessions. A total of 45 different distributions were generated, with AE increasing in 4° intervals from 0° to 180°. Bins with low AE (<4°) were chosen as the reference distribution, and compared against the other 44 for T11 (left panel) and T5 (right panel). The dotted line which represents the best linear regression fit, along with the Pearson correlation coefficients, $r$, is shown. **c** The reference distribution was estimated from neural features

(NF) time bins where AE < 4°, limited to day 0 where the decoder was first deployed for T11, and day 0 and 5 for T5. For subsequent sessions, neural distributions for comparison were constructed using an overlapping sliding window of 60 s at 1 s intervals. The KLD (right y-axis) is overlaid onto median AE calculated from the same sliding window (left y-axis in blue) across all recorded sessions for T11 (left panel) and T5 (right panel). Gray lines indicate the beginning of the sessions. Pearson and Spearman rank correlation, $r$ and $\rho$, respectively, quantify the relationship between the KLD and median AE. Insets present examples of cursor control of the task in the first and last session. For T11, cursor trajectories for all trials during a 5-min block are shown. Each color represents a peripheral target in a center-out-and-back task. For T5, cursor trajectories of the first 20 trials of a block are shown, along with the corresponding target presented at a random location on the screen in each trial.

only on the (marginal) distribution of recorded neural activity is much more widely applicable.

Here, we present and validate an approach to predict closed-loop decoding performance without the knowledge of true movement intention. MINDFUL was applied on longitudinal datasets where performance changed over long time periods as two people with tetraplegia, designated as T11 and T5, were using an iBCI. Each participant used an iBCI to control a computer cursor to perform target acquisition tasks on a screen. Target acquisition tasks permit observation of (presumed) motor intention and, hence, can be used to directly measure decoding performance. The kinematic decoder used by each participant was held fixed across all sessions so that persistent changes in performance could be ascribed to model drift and not to changes associated with the decoder. Briefly, MINDFUL represents changes in neural distributions relative to a reference distribution where the decoder was initially applied. MINDFUL is based solely on the recorded neural activity, without requiring information about the target locations. The resulting MINDFUL score was highly correlated with changes in closed-loop cursor performance over time.

## Results
### Fixed decoders result in initially stable and then unstable performance across months
To first establish a baseline for decoder performance, we deployed fixed decoders[27,51] for the purpose of identifying, over a comparatively long period, how neural instabilities may lead to deteriorating control. Data were collected from 15 research sessions, each from a separate day that spanned across 142 days, of T11 performing a center-out-and-back task using a fixed RNN (recurrent neural network) decoder, as previously described[51] (see "Methods"). As part of another study[27], T5 performed a random-target task for six sessions spanning 28 days using a fixed linear decoder (see "Methods"). To quantify closed-loop cursor performance, *angle error* (AE) between the inferred intended directional vector (cursor-to-target position) and the decoded velocity vector was used (see "Methods"). AE is a valuable metric for capturing performance as it is sensitive to instantaneous cursor direction change, and can be averaged across any range of time. T11 achieved stable, high-performance online cursor control for the first three months. The median AE per trial for T11 for sessions during the first three months was lower than later sessions on average (trial day 658–751: 26.8° ± 22.6°; trial day 758–800: 88.4° ± 46.1°; $p < 0.001$; Wilcoxon rank sum). For T5, the first three sessions demonstrated lower AE than the later three sessions (trial day 2121–2128: 39.6° ± 23.9°; trial day 2133–2149: 58.8° ± 31.7°; $p < 0.001$, Wilcoxon rank sum). Brief recovery from a decrease in performance was observed in both participants (93 days after the initial session for T11 and 28 days after the initial session for T5), indicating fixed decoders may not necessarily result in a steady decline in cursor control over time (Supplementary Fig. 1).

### Comparing distributions of neural activity patterns
MINDFUL is based on comparing the distribution of recorded neural activity patterns in a target dataset (usually with unknown decoding performance) to a similar distribution in a reference dataset (usually with known and good decoding performance). Figure 1a provides an illustration. The choice of neural features and measures of statistical dissimilarity are important for practical use. Here we used a measure of statistical dissimilarity based on the well-known Kullback–Leibler divergence (KLD). In principle, other measures of dissimilarity could be used (see "Methods"). Neural features were all derived from the inputs to the kinematic decoder: threshold-crossing spike rate and spike power for T11, and spike rate only for T5, extracted in 20 ms non-overlapping bins (see "Methods").

### Statistical distance between neural activity patterns correlates with performance
Having established datasets where fixed decoders result in periods of both stable and fluctuating closed-loop performance, we first investigated the underlying premise of MINDFUL that the distribution of neural activity

patterns varies systematically with decoder performance. First, neural features pooled from all sessions were categorized into groups based on performance in terms of instantaneous (20 ms) AE. KLD was computed to assess the differences in neural feature distribution at instances with low AE (<4°) to other distributions at instances with varying levels of AE (see "Methods"). For both participants, the relationship between the KLD and performance was found to be remarkably linear and strongly correlated (T11: Pearson $r = 0.985$, $p < 10^{-33}$; T5: Pearson $r = 0.983$, $p < 10^{-31}$; see Fig. 1b). Neural feature distributions at instances with low AE demonstrated high similarity (lower KLD) to the reference distribution of neural features at instances with low AE, and the KLD increased linearly as the compared neural feature distributions were drawn from instances with larger AE.

This is a proof-of-concept that statistical distance between distributions of neural activity patterns can correlate strongly with decoding performance. It does not, however, provide a measure of performance that would be useful in a clinical setting for detecting persistent changes in decoding performance that might arise from model drift. Instantaneous AE must be known a priori to define the collections of neural features that are compared in each point in Fig. 1b and MINDFUL is designed to be used in situations where AE is not known, at least not for the target distribution. Moreover, AE can be a result of noise (transient variability) or model drift (persistent changes), or both. Figure 1b does not distinguish among these even though model drift is the phenomenon of interest here. The linear relationship observed in Fig. 1b can be recreated in simulation using noise or using model drift or using both (see Supplementary Fig. 2).

### MINDFUL correlates with decoding performance
Towards developing a predictor of decoder performance based on neural activity for online applications, we define a measure called the MINDFUL score to study the effect of drifts that persist over timescales relevant to continuous iBCI use. Using the same concept as illustrated in Fig. 1a, but instead of grouping by AE as in Fig. 1b, neural feature distributions were estimated from collections of time bins aggregated using a 60-s sliding window, regardless of the performance during that window. The reference distribution is also estimated from instances of low AE as in Fig. 1b, but it is sub-selected from only the initial session(s) when the decoder is first deployed. As time progresses, we update the MINDFUL score which is based on the KLD between the reference distribution and the subsequent neural feature distributions from the sliding window (see "Methods"). To validate this method, the MINDFUL score is correlated against the median AE calculated in the same 60-s sliding intervals as in estimating the neural distributions. Strong correlations were found between the MINDFUL score and the median AE across sessions, especially for T11 (see Fig. 1c. T11: Pearson $r = 0.91$, $p < 0.001$; Spearman $\rho = 0.90$, $p < 0.001$; T5: Pearson $r = 0.59$, $p < 0.001$; Spearman $\rho = 0.60$, $p < 0.001$; see "Methods"). This suggests that MINDFUL can be a viable measure for tracking performance in real-time, as the statistical properties of neural features aggregated over a longer timescale window reflect information about how the decoder performance drifts over time without needing to know performance.

**Correlation to performance increases by combining neural features and decoder outputs.** The neural features (NF) in Fig. 1b, c were derived from principal components analysis (PCA; see "Methods") and need not reflect the sources of variability most predictive of decoder performance. Using features more closely related to the decoder output might strengthen the relationship between the MINDFUL score and AE. One such feature is the output from the decoder itself—in this case, the predicted 2-dimensional velocity vector, $\hat{X}$. Note that $\hat{X}$ alone cannot be used to define AE. It is the relationship between $\hat{X}$ and the true intended direction that defines AE. Nevertheless, changes in the distribution of $\hat{X}$ over a time interval may reflect changes in decoder performance. Same as Fig. 1c, reference distribution was estimated from sub-selected time bins of low AE from initial session(s), and target distributions were estimated from a 60-s sliding window. Except we used features from $\hat{X}$ instead of NF. Using only the 2-dimensional feature $\hat{X}$ showed reduced correlations

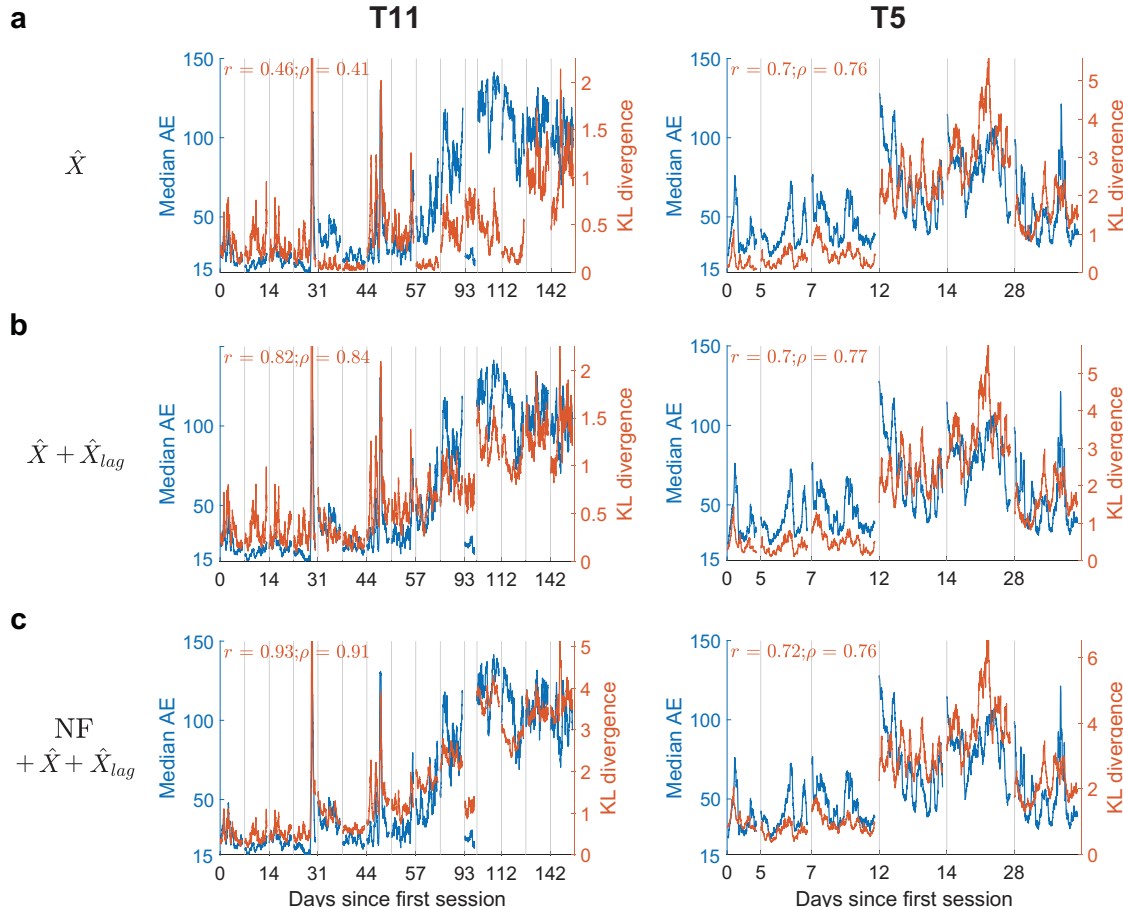

**Fig. 2 | Incorporating decoder outputs in the MINDFUL score maintains a high correlation with performance. a** The KLD (right $y$-axis) between distributions of decoded directional vectors, $\hat{X}$, with respect to the sub-selected time bins from the first session(s) overlaid onto median AE (left $y$-axis in blue) across all recorded sessions for T11 (left panel) and T5 (right panel). Subsequent neural distributions and median AE were updated every 1 s over a 60-s sliding window. Pearson $r$, and Spearman rank correlation coefficients $\rho$, between KLD and median AE are shown. **b** The KLD between distributions of $\hat{X}$ and $\hat{X}_{lag}$ overlaid onto median AE. **c** The KLD between distributions of the combination of derived neural features (as shown in Fig. 1c), decoded directional vectors, $\hat{X}$, and $\hat{X}_{lag}$, overlaid onto median AE.

between the MINDFUL score and AE than from just NF for T11, but increased for T5 (Fig. 2a. T11: Pearson $r = 0.464$, $p < 0.001$; Spearman $\rho = 0.407$, $p < 0.001$; T5: Pearson $r = 0.704$, $p < 0.001$; Spearman $\rho = 0.763$, $p < 0.001$). When using the 4-dimensional feature created by concatenating $\hat{X}$ and $\hat{X}_{lag}$, where $\hat{X}_{lag}$ comes from the previous time bin of $\hat{X}$ (20 ms earlier), there was an increased correlation to AE for both T11 and T5 (Fig. 2b. T11: Pearson $r = 0.819$, $p < 0.001$; Spearman $\rho = 0.840$, $p < 0.001$; T5: Pearson $r = 0.702$, $p < 0.001$; Spearman $\rho = 0.765$, $p < 0.001$). Lastly, the combination of inputs and outputs of the decoder, i.e. the low-dimensional NF, $\hat{X}$, and $\hat{X}_{lag}$ resulted in the highest correlation between KLD and AE for both participants (Fig. 2c. T11: Pearson $r = 0.926$, $p < 0.001$, Spearman $\rho = 0.913$, $p < 0.001$, T5: Pearson $r = 0.719$, $p < 0.001$, Spearman $\rho = 0.759$, $p < 0.001$).

**The MINDFUL score reflects changes in feature tuning.** We next investigated the types of changes in neural data captured by the MINDFUL score. Changes in directional tuning have been shown to reduce performance in both online and offline BCI studies[10,12,18–22]. Directional tuning was quantified by fitting a cosine function to normalized neural features to obtain estimates of preferred direction (PD) and modulation depth (MD)[33,55–57]. 154 out of 384 features and 85 out of 192 features, for T11 and T5 respectively, had significant directional tuning for at least half of all recording sessions ($F$-test, $p < 0.05$, see "Methods"). Changes in tuning in these features were tracked over time (see "Methods"). (Here we focused on the tuning of *individual* features, but we also used a different approach to show that changes in the

conditional distribution of *population* activity given motor intention are statistically significant; see Supplementary Fig. 3.)

Tuning properties shifted gradually for the majority of features in T11 (Fig. 3a, b). 125 out of 151 tuned features exhibited significant change in both MD and PD[35] in at least one session (see "Methods"). Fitted tuning curves across sessions for two example features illustrated changes in modulation depth and modulation lost, respectively for T11 (Fig. 3c). The average change in PD across these features was larger on days where performance was worse (day 7–93: 46.8° ± 31.2°; day 100–142: 62.4° ± 34.3°; $p < 10^{-7}$ Wilcoxon rank sum). The average absolute change in MD in later sessions was also found to be significantly larger (day 7–93: 0.107 ± 0.067; day 100–142: 0.159 ± 0.129; $p < 10^{-8}$, Wilcoxon rank sum).

Similar to T11, gradual changes in T5's tuning properties were also observed (Fig. 3d, e). Some features illustrated changes in either MD or PD, or both (Fig. 3f). 71 out of 85 tuned features exhibited significant change in both MD and PD in at least one session. The average change in PD across these features was larger on days where performance was worse (day 12 and 14: 69.9° ± 40.7°; day 5, 7, and 28: 58.1° ± 36.6°; $p = 0.0346$, Wilcoxon rank sum). Average absolute change in MD was not found to be significant (day 12 and 14: 0.147 ± 0.099; day 5, 7, and 28: 0.139 ± 0.093; $p = 0.664$, Wilcoxon rank sum).

To quantify the changes in encoding on a population level, we use tuning maps[56,58], defined by matrices of fitted tuning parameters of significantly tuned features on each session. Tuning similarity between days was assessed by calculating the correlation between the corresponding tuning maps (see "Methods"). In general, nearby sessions in time with

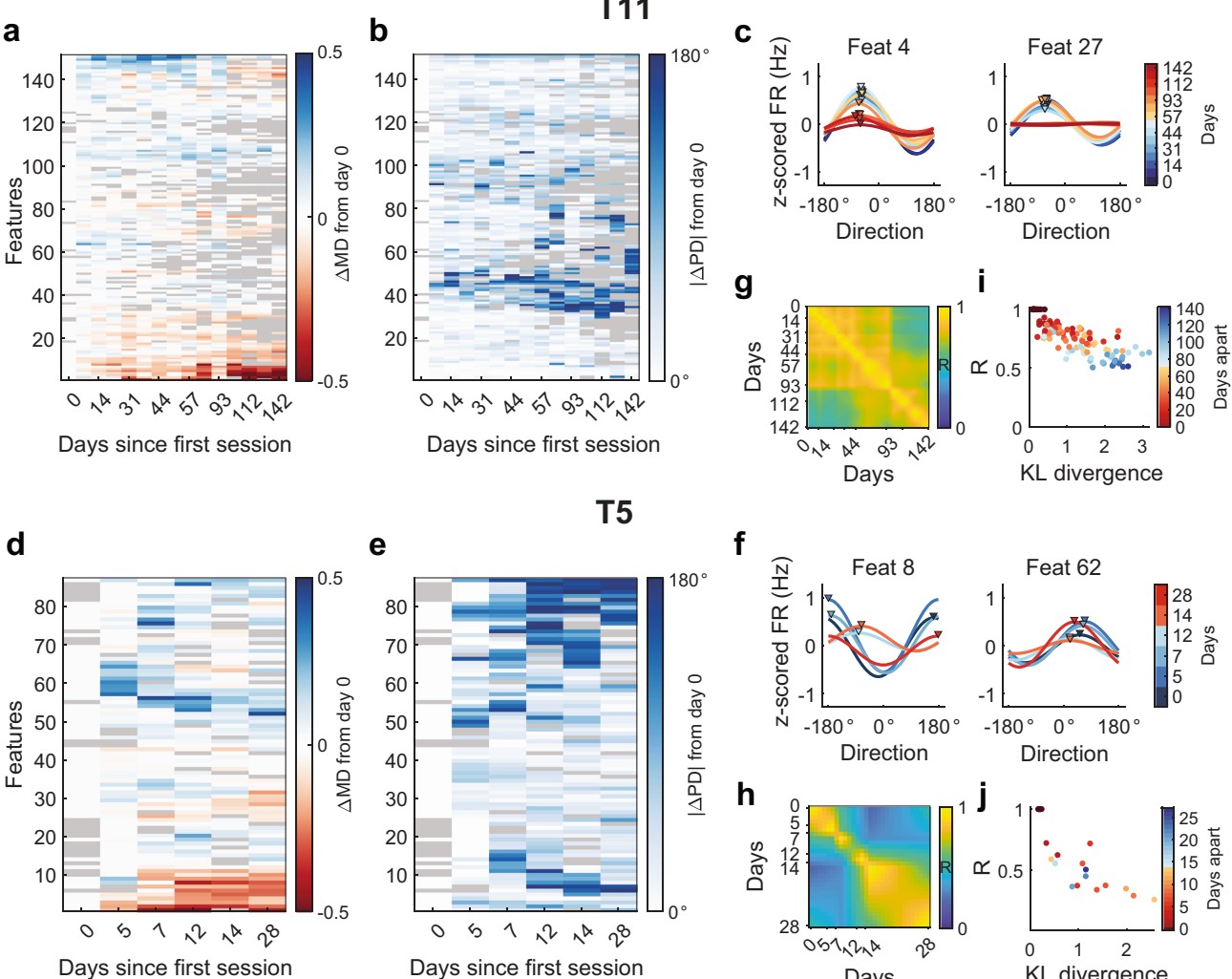

**Fig. 3 | Changes in feature tunings across sessions correlate with KLD. a** Changes in preferred directions (PD) and **b** modulation depth (MD) of significantly tuned features used in the decoder (relative to the tuning of the first day for which the feature was significantly tuned) for T11. Features were ordered by hierarchical clustering to visualize groups of features with similar tuning changes behavior (see "Methods"). Gray color indicates features that were not significantly tuned in that session. **c** Fitted cosine tuning curves for sample units across days for T11 illustrating changes in MD and channel dropout, respectively (color curves); Triangle markers denote PDs for sessions with significant tuning. **d** Changes in cosine tuning PD and **e** MD for significantly tuned features used in the decoder for T5. **f** Fitted cosine tuning curves for sample units across days of T5 illustrating changes in MD and PD, respectively. **g** T11 Tuning similarity across days represented by interpolated Pearson correlations between pairs of tuning maps (see "Methods"). **h** T5 tuning similarity across session days. **i** T11 mean KLD of neural distributions between sessions negatively correlates with the tuning similarity (Pearson $r = -0.812$, $p < 10^{-30}$, see "Methods"). Each dot corresponds to a pair of sessions with the color indicating the number of days apart. **j** T5 mean KLD of neural distributions between days negatively correlates with the tuning similarity (Pearson $r = -0.776$, $p < 10^{-4}$).

similar performance were more correlated. For T11, tuning maps among early sessions (up to day 93) with high performance were highly correlated, as well as among later sessions with low performance, but not across these two epochs (Fig. 3g). For T5, sessions that were closer together in time (along the diagonal) had higher correlation than those further apart (Fig. 3h). These results suggest that model drift (tuning changes) occurred across sessions. We were thus interested in determining how the MINDFUL score reflects changes in tuning similarity. To compare tuning similarity between sessions, we obtain a mean KLD between each pair of sessions. Instead of fixing a reference distribution, pairwise KLDs of neural features between sessions were calculated using a sliding window of 60 s updating every 10 s (see "Methods"). The KLDs from the same session were averaged to get a mean of the neural distribution difference between pairs of sessions.

For T11, pairs of sessions closer in time had smaller distribution shifts in terms of mean KLD, while pairs of sessions further from each other in time had larger distribution shifts, which consequently strongly correlated with

tuning similarity (Pearson $r = -0.812$, $p < 10^{-30}$, Fig. 3i). For T5, the same trend was observed, except for the pairs of sessions which compared the first three sessions to the last session where cursor control had recovered (dots in blue shades). The correlation between the mean KLD and tuning similarity was also strong and significant ($r = -0.776$, $p < 10^{-4}$, Fig. 3j). Together, these findings suggested that the MINDFUL score using KLD captures day-to-day changes in directional tuning, even though the metric can be calculated without information reflecting target position or movement intentions.

**The MINDFUL score captures low-dimensional neural latent space drifts.** We further investigated how MINDFUL relates to the changes in the low-dimensional neural latent space using demixed principal component analysis[59]. The top two direction-dependent principal components (PCs) on the neural population from decoder day 0 were calculated to compare changes across sessions by projecting the neural population from subsequent sessions on this PC space (see

**a**

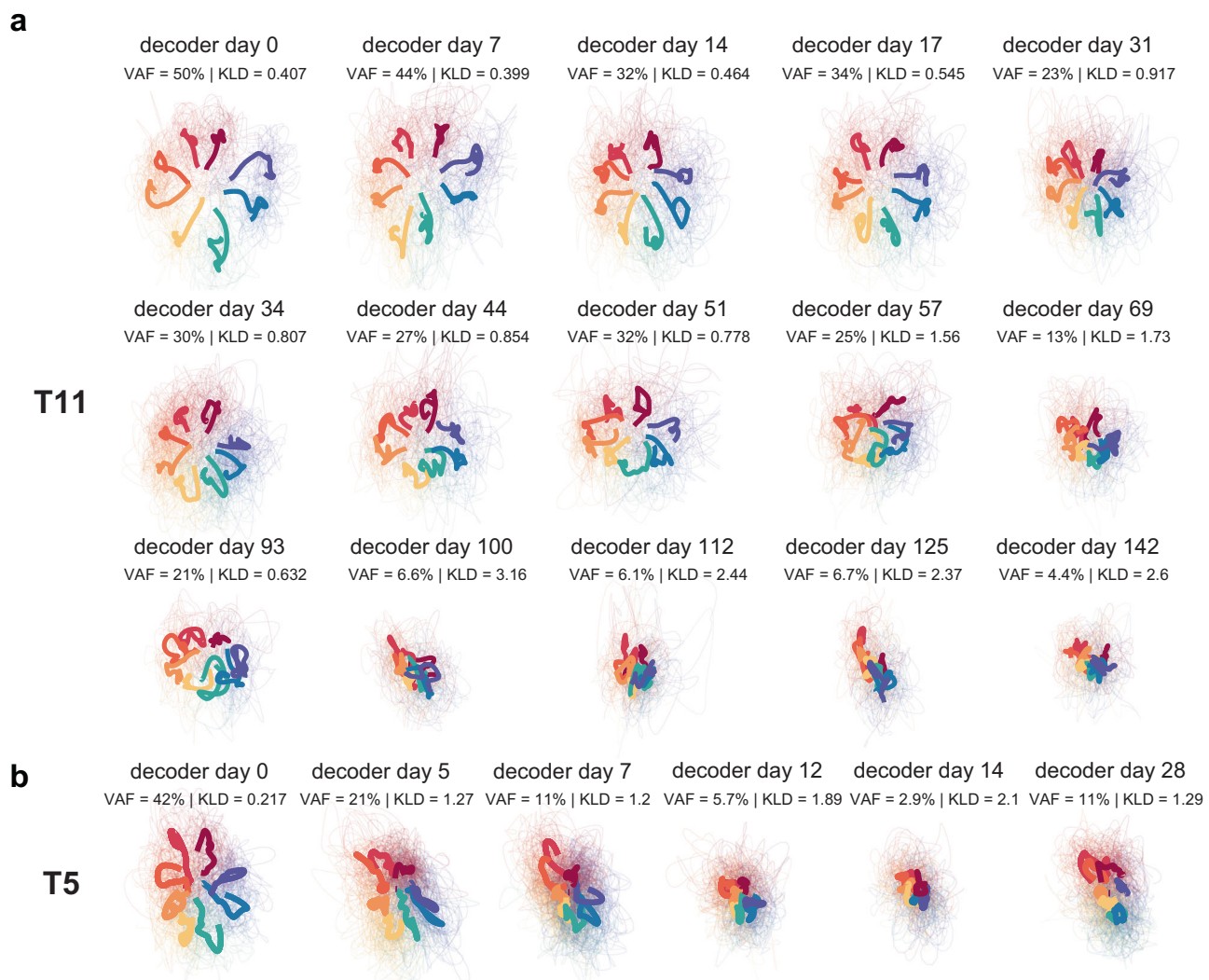

**Fig. 4 | Instability reflected in neural latent space. a** Projection of neural features of subsequent sessions onto the top two task-dependent PCs latent space of neural features on decoder day 0 using dPCA. Fine lines are trial trajectories and bold lines are trial averages per goal directions. Different colors correspond to the goal direction. **b** Projection of neural features for T5. For comparison simplicity, the random-target task was visualized and colored by discretizing the goal directions of each trial into eight even movement directions as in a center-out-and-back task.

"Methods"). The average neural trajectories per target directions became less distinct as the session days progressed, reflecting changes in the underlying population activity over time consistent with the decline in task performance (see Fig. 4). The amount of direction-related neural activity in each session was quantified by the variance accounted for (VAF) by the top two direction-dependent components on the subspace of day 0. For T11, the VAF was initially 50.0% on day 0 and remained above 20% on days in which clear separation of target trajectories was observed. As the decoder performance declined, the VAF dropped to 4.4% on decoder day 142 (Fig. 4a). This change in neural representation in low-dimensional space is strongly and significantly correlated to the mean KLD per session (Pearson $r = -0.892$, $p < 10^{-5}$). The mean KLD between day 0 and other sessions is calculated the same way as the mean KLD in the previous section (averaged KLDs using a sliding window of 60 s updating every 10 s, see "Methods"). For T5, VAF was initially 42.2% on day 0, and subsequently dropped to 2.9% and recovered to 11.3% on the last session (Fig. 4b). There was also a strong and significant correlation between the top 2 VAF and mean KLD (Pearson $r = -0.858$, $p = 0.0029$).

In addition to correlating with model drift in neural data, MINDFUL was found to detect large momentary deviations in the signal, likely

attributable to device-related reasons such as signal transmission errors[5]. The sharp spikes in both KLD and median AE in T11 neural data (Fig. 2) correspond to time bins during outlier trials (see Supplementary Fig. 4). Outlier trials were defined as having more than a 5% drop of wireless neural data packets or large "neural" responses greater than 8 standard deviations from the mean. Furthermore, when excluding these trials, the MINDFUL score was still highly correlated to the median AE (see Supplementary Fig. 5). This suggests that our method is capable of tracking both model drift over time, as well as short timescale technical-related variability. Although instantaneous events are less relevant for decoder recalibration, a method to capture these events may prove useful in other iBCI troubleshooting with both the current and future fully implanted systems.

**Selecting reference and window length further optimize correlation.**
We explored the role of sub-selecting time bins with different AE ranges as the reference in the **MINDFUL** pipeline. When limiting the reference to the collection of time bins with low AE only (0–4°) as shown in Figs. 1c and 2, there are strong correlations between the KLD of derived neural features (NF + $\hat{X}$ + $\hat{X}_{lag}$) to AE. It was higher than when using all-time bins of any cursor control quality for both participants (see Fig. 5a). This also held true when taking other combinations of derived neural

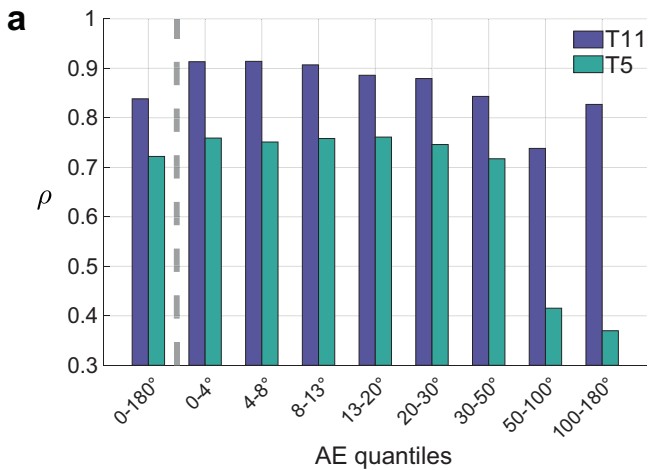

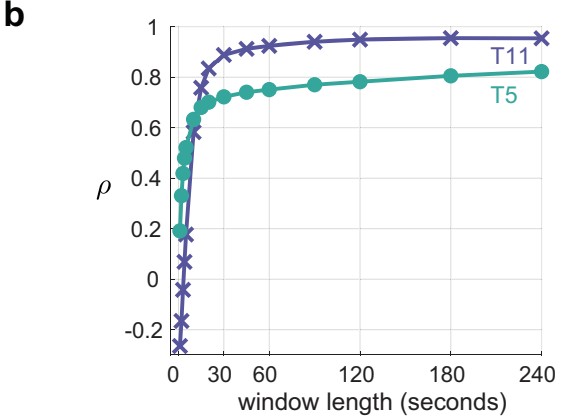

**Fig. 5 | Sub-selecting instances of low AE for reference and using longer window length improve correlation between MINDFUL and AE. a** Spearman correlation coefficients of KLD of NF, $\hat{X}$, and $\hat{X}_{lag}$, to AE when sub-selecting time bins with different quantiles of AE as the reference. Each quantile approximately contains an even number of observations. **b** Spearman correlation coefficients of KLD to AE when using different window lengths to estimate target distributions.

features into calculating KLD (Supplementary Table 1). In addition, sub-selecting instances with high AE (50°–100°, 100°–180°) as the reference distribution reduced the KLD-AE correlation for both participants, especially in T5.

Another important consideration for improving the correlation between MINDFUL and performance is the duration of neural data required to obtain a reasonable representation of neural distribution. Since neural activity recorded in the precentral gyrus modulates with the direction of intended movement, too small of a window length may reflect task-dependent differences in reaching different directions, rather than a persistent model drift that affects cursor control regardless of trial direction. A longer window length can avoid this issue when directional distribution differences are averaged out over a longer period. However, the KLD would be smoothed out if the window length is too long, resulting in a larger delay to detect the need to update the decoder. To balance between accuracy and efficiency for online implementation, the optimal duration was empirically determined to be at least 60 s (Fig. 5b), where the KLD-AE correlation began to plateau, and longer windows did not offer a higher correlation. Using a 60-s window to estimate neural distributions provides a sufficiently large number of samples to average out directional-dependent differences due to variations in trial-to-trial movement directions.

**MINDFUL is robust to the reference task**. We repeated the analysis of tracking the correlation of MINDFUL to performance (see Fig. 2c) for T11 except we used reference data collected during different tasks. The

comparison data collected during center-out-and-back tasks remains the same as in Fig. 2c. Since the relationship between neural activity and movement can be context and task-dependent[60,61], it is unclear to what degree the reference and comparison tasks must be matched for MINDFUL to correlate well to performance. MINDFUL is likely to be most useful in practice if it is robust across tasks and contexts. Collecting reference data from a different task was not part of our original experimental design, but for T11 the appropriate data was collected for other purposes. In addition to the center-out-and-back task, T11 used the same fixed decoder for random-target tasks (day 7), analogous to T5's task described above, and during personal iBCI use (i.e., browsing the web; day 0; see "Methods"). Despite different cursor tasks being used as the reference from the target distributions, MINDFUL still correlates highly to performance, even without the help of subsampling based on AE (see Fig. 6b, c). This is also true when all three types of tasks were combined together to estimate the reference distribution (mixed tasks, see Fig. 6d). The range of KLD is slightly higher for the random-target task, and lower for personal use and mixed tasks. This suggests that MINDFUL might be robust to cursor task changes for reference.

## Discussion

Apparent model drift during chronic iBCI use—resulting from changes in the information encoded in neural ensembles, changes in the recorded neural elements themselves, or changes in the recording devices—is one of the major challenges for developing decoders that will provide stable, accurate cursor control for long-term use by people with paralysis. Existing solutions to mitigate more substantial model drift have limitations. Explicit recalibration comes at the expense of interrupting the user in the midst of iBCI use to collect additional data for decoder training. While background recalibration using self-supervised machine learning algorithms doesn't require the user to perform daily repetition of a task, it relies on stable online performance for effective pseudo-labeling. Currently, one cannot predict accurately the moment at which the decoder may fail to sustain performance and thus require a supervised recalibration. Similarly, robust algorithms reduce the need for frequent retraining but may require retraining of the model de novo. MINDFUL fills in the gap in the development of a better decoder recalibration strategy for practical everyday use by identifying, quantifying, and monitoring the degree of neural recording instability that contributes to the degradation of real-time decoder performance.

The MINDFUL score, which is based on measuring the Kullback–Leibler divergence (KLD) between distributions of neural features, reflects online performance without the need to incorporate knowledge of intended targets. In two participants, the MINDFUL score was strongly correlated to online angle error during iBCI cursor control across session days spanning up to four months (T11) or one month (T5). With the goal of translating this method to an online setting for the purpose of personal iBCI use, the MINDFUL score can reflect performance accurately for different cursor tasks examined in different participants (Fig. 2). Importantly, the MINDFUL score is consistent with tuning and latent space changes which cannot be directly measured without information about movement intention. This suggests that the MINDFUL score provides an intrinsic measure to track model drift affecting decoder performance during long-term iBCI use.

Our study confirmed the well-acknowledged observation that model drift can impact online performance when the decoder cannot accommodate neural changes over long-term iBCI use. Model drift was quantified by tracking changes in tuning and latent space representations of neural population activity across sessions. The MINDFUL score was found to be highly correlated with both of these measures. It should be noted that our method did not track mean firing rate shifts which are known to correlate with declines in decoder performance[19]. In our datasets, adaptive mean corrections such as $z$-scoring or bias correction were applied to the neural features during online cursor control to combat this type of model drift (see "Methods"). Therefore, performance drops observed in this dataset were largely due to other types of model drift. The MINDFUL score, which

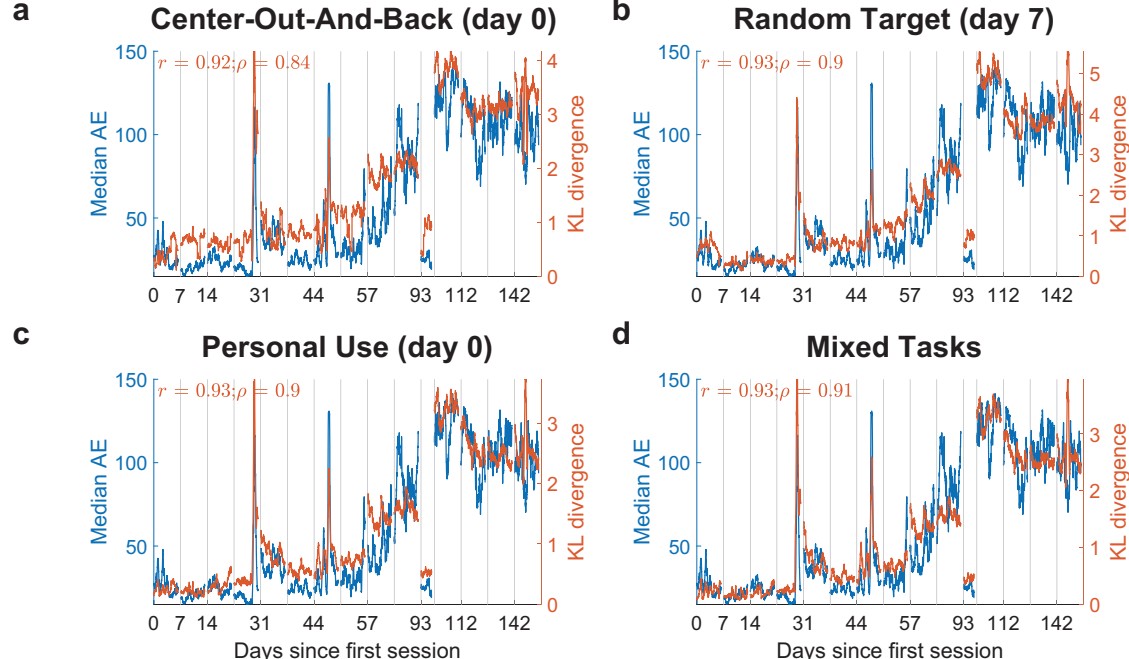

**Fig. 6 | MINDFUL reference can be applied across multiple cursor control tasks.** Instead of using data from day 0 where participant T11 was performing the same tasks in subsequent days (center-out-and-back), different 2D cursor tasks were selected as reference. In these tasks, T11 used the same fixed RNN decoder. The same PCA procedure as described in the "Methods" was applied. No subsampling on angle error was performed on the reference as it is not explicitly available during personal use. Features used include all neural features and decoded kinematics plus its lagged version. **a** Reference was estimated from a center-out-and-back task on day 0, same as in Fig. 2c except time bins were not sub-selected to have AE less than 4°. **b** Reference was estimated from T11 performing a random-target task for 10 minutes on day 7, using the same fixed decoder. Task setup is very similar to T5's data except with varying target sizes. T11 did not perform this task on day 0. **c** Reference was estimated from when T11 was using the iBCI for personal desktop use, such as browsing the internet, on day 0. Around 16 minutes of active cursor control period was included. **d** Reference was set to a combination of the above-mentioned data (concatenating random-target task, personal use, and center-out-and-back task).

measures changes in the distribution of mean-corrected neural data, aims to discover model drift such as changes in tuning or latent representation which may necessitate a decoder recalibration to restore control. Furthermore, since MINDFUL was designed to be applied during online iBCI control where threshold crossings were primarily used as neural features[62–65], we chose to investigate the functional stability in the thresholded neural activity' tuning properties and latent neural representation rather than neuronal stability of discriminated single units processed via spike sorting techniques[18,31].

Another plausible contributor to the observed changes in neural distributions may be participants' compensatory neuro-motor strategies in response to suboptimal cursor control. Non-human primates that encounter artificial perturbations in a previously learned BCI motor control task elicit new neural patterns with learning[54,66–68]. Neural activity during closed-loop, online control contexts is also different from open-loop, offline control which has added real-time visual feedback[69,70]. In this closed-loop iBCI study, when participants experienced a directional bias in cursor kinematics, it is plausible that the participants might compensate for decoding errors with different strategies such as moving against the bias (eliciting a larger magnitude of velocity), temporarily pausing attempting movement control, or moving towards the bias in hope to reset the bias (when automatic bias correction is applied). These alternative strategies are valid responses in attempting to improve control, but they may result in a larger change in the distribution of neural features, amplifying the original model drift when the intention context remains consistent. This highlights one of the challenges when studying model drift during closed-loop control when the ground-truth intended movement cannot be observed independently of the decoded outputs.

While the MINDFUL score based on KLD consistently correlates with performance and changes in neural representation, there are a number of noticeable differences in the results between subjects. First, the relevance of

the chosen neural features appears to be different for different participants. For example, for participant T11, the MINDFUL score derived from the neural features is more strongly correlated to the AE than the MINDFUL score derived from decoded velocity features, slightly improving when combining both features; for participant T5, the MINDFUL score derived from the decoded velocity features is more strongly correlated to the AE than neural features alone and is not substantially improved by adding the neural features (Figs. 1c and 2). The features used, choice of decoder and cursor task, and duration of data collection, can all influence the value of KLD, hence correlation to AE. Second, there are noticeable differences in the range of KLD between subjects ([0, ~2.75] for T11 and [0, ~0.9] for T5 in Fig. 1c). It is possible that variability between subjects and between the dataset may affect the range of KLD in MINDFUL. Interestingly, the range of KLD calculated by binning by performance is much smaller than calculated across time ([0, ~0.3] for both participants; Fig. 1b). In Fig. 1b, since data was collected across a wider range of time, both for the reference and comparison distributions, any model drift would likely cause these distributions to have larger variance and, hence, smaller KLD. In support of this conjecture, we found that the determinant of the empirical covariance matrix of the reference distribution by binning by performance (Fig. 1b) is 3.4 times larger than the reference set to the first session (Fig. 1c) for T11, whilst for T5, these determinants are relatively equal. Despite a number of methodological differences between the datasets and chosen features, it is encouraging that MINDFUL robustly measured model drift for both participants.

Choosing an appropriate reference when calculating KLD influences the reliability of predicting decoder performance using MINDFUL. First, selecting low AE time steps for reference was found to provide a higher correlation between the KLD and AE (Fig. 5a). Using low AE as a reference helps to identify the model drift where the neural-kinematics distribution during decoder training has changed from that during testing when the

decoder was applied online. The training data of the decoder typically represents periods of relatively high performance: For T11, the fixed RNN decoder was trained on selected historical data with angle error <45°, while T5's decoder was initially trained on open-loop blocks and then immediately updated with a closed-loop block[27,51]. As neural representations shift from the training distribution, the decoder is more likely to produce a subpar performance with a higher error rate. Therefore, when sub-selecting only high-performance data as the baseline, future neural shifts can be more accurately reflected by the KLD. However, there exists subject variability and ambiguity regarding precisely how much data is needed for reference. For example, for participant T5, using time bins with high AE (50°–100° or 100°–180°) as reference resulted in a more drastic decrease in the correlation between KLD and AE than T11. Also, using the first two sessions as a reference resulted in a slightly higher Pearson correlation than just the first session alone for T5 (Supplementary Fig. 6). Nevertheless, assuming that a newly trained decoder returns decent initial performance, our findings suggested that data from high-performance time bins from the initial session where the decoder was first applied would be an appropriate choice for the reference.

In this study, the MINDFUL score based on neural activity alone reflects performance in cases where fixed decoders were used online. However, it remains unclear how it can be applied to other types of adaptive or robust decoders that aim to stabilize decoding by periodically realigning neural data to the initial session. In such circumstances, if MINDFUL is calculated before data alignment, it may not directly correlate to performance as adaptive alignment may keep AE low even when neural representations are changing. However, MINDFUL could still be useful in several ways. First, MINDFUL may be applied to transformed neural data after manifold alignment methods. Even if alignment approaches will result in a reduction in KLD (KLD is a common choice of loss function), MINDFUL can measure the remaining differences. Second, if the features used in MINDFUL are the outputs from the adapted decoder, rather than non-adapted features such as PCA components, then MINDFUL might continue to correlate to online performance despite decoder adaptation. Lastly, instead of periodically aligning (or recalibrating), MINDFUL can be used to signal the need for recalibration when model drift is detected. Future experiments with other closed-loop adaptive decoders will be required to test these approaches.

We proposed a statistical method to detect model drift when fixed decoders were used during consecutive days-to-months of iBCI cursor control by two people with tetraplegia. This is crucial for the goal of clinical translation of iBCI systems for practical everyday use, as it requires stable and reliable decoders to maintain high performance despite drifts in neural representations over time. MINDFUL was shown to be able to track model drift based on the intrinsic properties of neural features and decoder outputs, which correlates to long-term changes in decoder performance, without needing to be aware of the movement intention. This approach is well-suited for future online adaptive iBCI systems aiming to provide continuous long-term control in a practical, personal setting outside of standardized research sessions, where it is not possible to directly track intended movements. For instance, MINDFUL and related methods could be used to trigger either a user-engaged or background update as the decoder becomes less effective.

There are additional several considerations for applying the MINDFUL score online. First, during personal iBCI use, cursor movement directions could be less symmetric and more sparsely distributed than the cursor tasks in this study. When sub-selecting reference time bins based on movement direction (up/down/left/right), KLD became higher in magnitude and generally less correlated to AE (see Supplementary Figs. 10 and 11). A longer time window or careful time bin selection for both reference and target distributions may be needed to reduce the directional-related differences. In addition, we believe that MINDFUL may be useful for detecting changes in the relationship between the signal and the decoder(s), even when multiple disparate tasks and contexts are incorporated into the reference distribution. For T11, MINDFUL was found to be robust to using neural reference data collected during different cursor tasks, including periods of personal iBCI use for which we had no control over the balance of intended directions or angle error. We did not have data to investigate this in T5. Future work will investigate this robustness in additional participants and more varied changes in tasks and contexts, all of which will be important for practical iBCI use.

Second, as previously described, the range of KLD varies between participants. It will be crucial to set an appropriate threshold for triggering a recalibration for individual users. One possible strategy to set a user-specific threshold would be to initialize a threshold based on an AE cut-off from previously collected datasets and iteratively fine-tune the threshold sensitivity by incorporating the user's feedback. Lastly, the above-mentioned large noise instances which can be easily detected by MINDFUL should not trigger a recalibration, as it does not imply a change in the neural-kinematic relationship estimated by the decoder. The frequency or pattern of these events could, however, inform further iBCI development.

## Methods

### Human participants
The Institutional Review Boards of Mass General Brigham/Massachusetts General Hospital, Brown University, Providence VA Medical Center, and Stanford University granted permission for this study. Intracortical neural signals were recorded from participant T11, a 37-year-old right-handed male with a C4 AIS-B spinal cord injury (SCI) that occurred approximately 11 years prior to study enrollment, and T5, a 65-year-old right-handed male, with a C4 AIS-C SCI that occurred approximately 9 years prior to study enrollment. Both participants are enrolled in the BrainGate2 pilot clinical trial (NCT00912041), permitted under an Investigational Device Exemption (IDE) by the US Food and Drug Administration (Investigational Device Exemption #G090003; CAUTION: Investigational device. Limited by Federal law to investigational use). Informed consent was obtained from all participants. All research sessions were performed at the participant's place of residence. All ethical regulations relevant to human research participants were followed.

### Intracortical neural recordings and neural features
Each participant had two 96-channel microelectrode arrays (Blackrock Neurotech, Salt Lake City, UT) placed in the dominant (left) hand/arm knob area of the precentral gyrus[2]. T11's intracortical neural signals were recorded via a wireless broadband iBCI system[71] while T5's neural signals were acquired via the cabled iBCI system. The average signal across the array per electrode was subtracted with a common average reference filter to reduce common mode noise. Neural features were extracted from the neural recording in 20 ms non-overlapping bins. For real-time decoding and off-line analysis, multi-unit threshold-crossing spike rates (RMS < −3.5) per electrode were used for T5, and two types of features: spike rates (RMS < −3.5) and power in the spike band (250–5000 Hz) per electrode were used for T11. Across the 15 T11's sessions, 34 of the 1840 trials were labeled as outlier trials, which were defined as having more than a 5% drop of wireless neural data packets or large "neural" responses greater than 8 standard deviations from the mean. No outlier trials were identified in T5's sessions.

### BCI behavioral task
To assess decoder performance for cursor trajectories, T11 performed a closed-loop 2D point-and-click center-out-and-back task for each of 15 sessions on separate days that spanned 4 months (trial days 658–800). For each trial, T11 was prompted to attempt hand or finger movements to continuously move the neural cursor from the center target to one of the eight pseudo-randomly selected peripheral targets and to then attempt a hand gesture (right index finger down) to click on the target. T11 was encouraged to maintain the same set of motor imagery for all sessions presented in this study. Upon target selection, in the next trial, T11 was asked to move the cursor back to the center target. A trial is successful when the cursor is inside the target and a click action is decoded. Otherwise, a trial is considered failed after a 10-s timeout. Each session consists of two 5-min

task blocks, except for trial day 751 with only one block. The cumulative task time of all sessions is 145 min, with a total of 1840 trials. Neural features, cursor position, target position, and decoder velocity outputs were logged.

T5 performed a closed-loop 2D random-target selection task with a fixed-size target appearing in random locations on the screen. T5 attempted to move the cursor over the target and dwell on it for a consecutive 500 ms period before the 10-s timeout to complete the trial. Audio feedback was provided right after the end of a trial to indicate trial success. A new random-target is immediately presented with no delay. This task was repeated for 6 sessions on separate days that spanned 28 days (trial days 2121–2149). Each session consists of two to four 4-min closed-loop blocks, which provide 84 min of 1200 total trials across all sessions. Training blocks for calibrating the decoder on trial day 2121 were not included in this study.

### Angle error
The instantaneous angle error is defined as the absolute angle difference between the inferred intended directional vector (cursor position to target position) and the decoded velocity vector, $\hat{X}$ (best = 0°; max = 180°). In each 60-s interval to estimate the neural distribution for calculating the KLD, the median of the angle error of time bins within the same interval is also computed. The median is chosen over the mean because AE in our datasets for both participants is not uniformly distributed between 0° and 180° (skew towards lower AE).

### Closed-loop neural decoding
Decoders in this study were previously described in refs. [27],[51]. Briefly, for T11, an LSTM decoder was used to infer movement intentions from neural recordings. An LSTM is a variant of recurrent neural network (RNN) with improved capability for long-term temporal dependencies[72]. Previous studies described the advantage of using an RNN for neural decoding over linear methods such as the Kalman filter[20],[73–75]. The LSTM decoder was trained and validated on closed-loop point-and-click cursor tasks from the 18 most recent sessions of T11, spanning 70 days from trial day 576 to 646. Of these sessions, only task blocks with a median angle error of less than 45° were included, which yielded a total of 331 minutes or 8441 trials of training data. Input neural features were passed directly to the RNN layer whose outputs went to three densely connected activation functions, decoding the x- and y-velocity and the distance to the target. During online control, each neural feature was adaptively z-scored using the mean and variance from a 3-minute rolling average window.

Clicks were decoded with a linear discriminant analysis (LDA) followed by a hidden Markov model. The LDA calculates a subspace that maximally discriminates between a click and a movement state. Coefficients were estimated with a regularization term of 0.001. Emission means and covariances used the empirical mean and covariance from the training data. The selected z-scored neural features were smoothed with a 100 ms boxcar window before being projected onto the LDA space. The estimated class probabilities were normalized using the SoftMax function and then smoothed with a 400 ms boxcar window. A click is returned when the click probability is above a threshold of 0.98.

For T5, non-overlapping 20 ms-binned extracted features were fed through a linear regression model trained to predict the cursor-to-target distance. An initial decoder was trained based on T5's neural activity while he engaged in an open-loop block on day 0 (trial day 2121). This decoder was then used to drive closed-loop control in a subsequent block. The final decoder parameters were then updated based on the first closed-loop block, and they were fixed for later closed-loop blocks and future sessions. The raw decoded velocity $v_t$ was exponentially smoothed with the running velocity average $\hat{X}_t$ via $\hat{X}_t = \alpha\hat{X}_{t-1} + (1 - \alpha)\beta v_t$, where $\alpha$ is the smoothing factor and $\beta$ is the gain parameter. Smoothing and gain were manually adjusted during the first session and fixed on subsequent days.

To accommodate for session-to-session variability in recordings, we applied per-channel z-scoring at every time bin for T11 and a bias correction for T5. For T11, mean and variance were initialized from the previous block and adaptively updated them using a 3-min rolling window. Neural features

were decoded into cursor velocities by a real-time LSTM decoder. For T5, a bias correction was applied to mitigate mean shifts in the decoded output by subtracting a running estimate of the decoder bias from the velocity outputs (with an adaption rate of 0.3)[11]. Bias correction was first initialized from the previous blocks. The intercept term in the decoder is then updated to the negative resulting bias vector (obtained by pushing the mean firing rate vector through the decoder weights).

### Derived neural features
MINDFUL can be based on any collection of neural features. In this paper, we experimented with three different types of features. The first is extracted neural features as described above (384 dimensions for T11 and 192 dimensions for T5). Individual neural features were z-scored per-channel using a 3-min rolling window as implemented during online iBCI control for T11. The same procedure is applied to T5 despite a bias correction approach being applied during online control to offset means drifts. The second is based on principal components analysis (PCA) of the extracted neural features (after z-scoring). The recorded neural features were projected onto the PCA subspace defined by the top M principal components (PCs) of a reference dataset that we call the *PCA-reference data*; see below. The third is based on the output of the decoder (which can be viewed as a type of neural feature), $\hat{X}$. We also consider $\hat{X}_{lag}$, the previous time bin (20 ms earlier) of $\hat{X}$.

### Kullback–Leibler divergence (KLD)
The MINDFUL score is based on comparing two datasets of neural features, defined in our case by the neural feature vectors from two collections of time bins. We use the derived neural features as described above. The first collection of time bins defines the reference *data $P_1$* and the second defines the *comparison data $P_2$*. Our choices for the reference and comparison time bins are described below. We first compute the sample mean (column) vectors $\mu_1$ and $\mu_2$ of the neural feature vectors in the reference data $P_1$ and comparison data $P_2$, respectively. These mean vectors are the same dimension k as the derived neural features. We similarly compute the $k \times k$ sample covariance matrices $\Sigma_1$ and $\Sigma_2$ in the two datasets. The MINDFUL score that we use is

$$d_{KL}(P_1||P_2) = \frac{1}{2}\left(\mathrm{tr}(\Sigma_2^{-1}\Sigma_1) + (\mu_2 - \mu_1)\Sigma_2^{-1}(\mu_2 - \mu_1) - k + \ln\left(\frac{\det\Sigma_2}{\det\Sigma_1}\right)\right)$$

where tr(·) and det(·) denote the trace and determinant of a matrix, respectively, and ln(·) is the natural logarithm. This formula is the Kullback–Leibler divergence (KLD) between two k-dimensional multivariate Gaussian distributions with respective mean vectors $\mu_1$ and $\mu_2$ and respective covariance matrices $\Sigma_1$ and $\Sigma_2$. Although it is motivated by a multivariate Gaussian model, its utility as a score for comparing two datasets does not rely on a Gaussian assumption. In developing MINDFUL, we experimented with other measures of statistical difference based only on means and covariances, such as the Jeffrey's (symmetric KL), Bhattacharyya, and Wasserstein distances between multivariate Gaussians, and found qualitatively similar results. We selected KLD as the example for this paper because it consistently gave the best results in many different scenarios, and it is widely known.

### KLD grouped by angle error
In Fig. 1b, the reference and comparison are grouped by the angle error (AE) of the time bin, regardless of movement intention and session day. The reference data consists of all time bins for which the AE < 4°. We used these same time bins for the PCA-reference data. The comparison data consists of time bins for which the AE is in a particular 4° interval. We used 45 different comparison datasets defined by the AE intervals [0°,4°), [4°,8°), …, [172°,176°), [176°,180°] giving 45 different KLD scores, calculated as described above. These scores are plotted versus AE (using the middle of each AE interval for the AE value) in Fig. 1b, and these 45 pairs define the reported correlations for Fig. 1b. The derived neural features used are the top $M = 5$ PCs.

## MINDFUL score to track model drift during closed-loop iBCI control

MINDFUL quantifies the neural distribution shifts over time relative to a historical reference distribution. We use KLD as described above and experiment with different choices of derived neural features and reference and comparison time bins. The reference time bins are restricted to the first session when the decoder was first deployed for T11, and the first two sessions for T5 (the first session is shorter than the others; see supplementary Fig. 6 for KLD using only the first session for reference). We additionally restricted the reference time bins to those with AE < 4° in Figs. 1c, 2, 5b. We varied this AE threshold for inclusion in the reference data in Fig. 5a. In all of these figures the PCA-reference data is the same as the reference data. In Figs. 1c, 2, 5a, the comparison data consists of all time bins in a 60-s interval. In Fig. 5b, we varied the length of the comparison data interval. In all of these figures, the comparison data interval is shifted in increments of 1 s to investigate how the MINDFUL score varies over time. In Fig. 1, the derived neural features are the top M = 5 PCs. In Fig. 2a, the derived features are the 2-dimensional decoder output $\hat{X}$. In Fig. 2b, the derived features are 4-dimensional and consist of $\hat{X}$ and the 2-dimensional decoder output from the previous time step $\hat{X}_{lag}$. In Figs. 2c and 5, the derived features are $\hat{X}$, $\hat{X}_{lag}$, and the top $M = 5$ PCs, for a total of 9 dimensions. Supplementary table 1 shows results with additional choices of the derived features (no $z$-score, $M = 10$, or no PCA).

## MINDFUL robustness to reference data across tasks

In Fig. 6, we repeated the process of calculating MINDFUL as in Fig. 2c except using different task data to estimate the reference distribution. Additionally, no subsampling based on AE was applied for the reference, as the ground-truth performance is not available during personal use (no target was cued and therefore performance metrics are not readily available). Approximately 16 min of personal use and 10 min of random-target tasks were included for reference. The target distributions were still estimated from a 60-s sliding window during the center-out-and-back task in subsequent sessions. Thus, median AEs calculated over time bins of these target distributions remain the same as Fig. 2c in this analysis.

## Cosine tuning

We fit cosine tuning curves to estimate the tuning properties per feature per session. Cosine tuning has been used to describe the relationship between the neuronal firing rate to movement directions, and it forms a basis for using a linear decoder for neural decoding[57]. In a cosine model $y = b_0 + b_1 \cos\theta + b_2 \sin\theta$, $y$, the firing rate of a neuron, is regressed on $\theta$, the movement direction. $b_0$, $b_1$, $b_2$ are regression coefficients that can be estimated with least squares unbiased estimators. The model can also be expressed equivalently as $y = b_0 + \alpha \cos(\theta - \theta_0)$, where $\alpha = \sqrt{b_1^2 + b_2^2}$ representing the modulation depth (MD) of the cosine curve. and $\theta_0$ representing the preferred direction (PD) where the largest firing rates are recorded. It is noted that since features in this study were $z$-scored, the bias term $b_0$ is closed to zero, and is therefore omitted in the calculation of modulation depth. Tuning parameters were estimated from 20 ms-binned features from the first second after the go-cue of non-outlier trials (offset by 160 ms reaction time) to capture neural activity associated with reach initiations. Feature tunings are considered significant if the $F$-test on the regression model has a $p$-value < 0.05. See Supplementary Fig. 7 for the estimated tuning and the empirical firing rate of example features.

## Changes in MD and PD

For each feature on each day, change in MD was calculated by $\Delta MD = MD - MD_{ref}$, and change in PD was calculated by $|\Delta PD| = \min(360° - |PD - PD_{ref}|, |PD - PD_{ref}|)$, where $MD_{ref}$ and $PD_{ref}$ refer to the tuning of the first day for which the feature was significantly tuned. The CircStat toolbox was used[76]. Only features that have significant directional tuning for more than half of all sessions were considered in the tuning map described below. (For all features including the

non-significantly tuned, see Supplementary Fig. 8). Significance of tuning change was assessed by bootstrapping samples of PD (or MD) to obtain a distribution of $\Delta MD$ and $\Delta PD$[33,35]. If the 95% confidence interval for the difference distribution does not contain 0, then we reject the null hypothesis at the 5% significance level[35]. To visualize the patterns of tuning changes, the features were ordered by their tuning parameters using hierarchical clustering on Matlab. Euclidean distance was used to estimate the similarity of standardized $\Delta MD$ and $\Delta PD$ from all sessions between two features, and ward linkage was applied to arrange the order of the clusters. The same ordering was used in the heatmap of $\Delta MD$ and $\Delta PD$ for the same participant.

## Changes in tuning map

We quantified changes in directional tuning on a population level by comparing tuning maps over recorded sessions. A tuning map on each session is a 3 x N matrix comprising the fitted tuning curve parameters, $b_0$, $b_1$, $b_2$, of $N$ number of features with significant tuning on that day. Pairwise Pearson correlations of maps were performed to assess the similarity of tuning across sessions. In a pair of daily tuning maps, only features that were significant on both maps were considered to calculate the correlation. This pairwise correlation was plotted in a heatmap which was interpolated to account for the irregular number of days apart between sessions.

## Mean KLD between sessions

To estimate the average neural distribution difference between pairs of sessions, a mean KLD between distributions on two given sessions was calculated (see Fig. 3e, j). The 9-dimensional derived neural features are the same as Figs. 2c and 5, namely, the top $M = 5$ PCs, $\hat{X}$, and $\hat{X}_{lag}$. The PCA-reference data is the same as Figs. 1c, 2, and 5, namely, time bins from the first session (T11) or the first two sessions (T5) that have AE < 4°. The reference and comparison data are each 60-s intervals updated every 10 s. If there are M such intervals in session $i$ and N in session $j$, then the mean KLD between the sessions is the average KLD of all $(M \times N)$ pairs of intervals from sessions $i$ and $j$. Outlier trials were excluded from the intervals. For visualizing the complete pairwise comparison matrix containing the KLD of all pairs of intervals from all sessions, see Supplementary Fig. 9.

## Latent space dPCA projection

We applied demixed principal component analysis (dPCA)[59] to population neural activity from the initial decoder day for T11 and T5 respectively. For subsequent trial days, we projected neural data onto the top two task-relevant neural dimensions dPCA space of day 0. PCs were computed from all features of the first second after the go-cue of non-outlier trials (offset by 160 ms reaction time). Neural features were smoothed using a Gaussian kernel with a standard deviation of 50 ms. T5 random-target task trials were discretized into eight movement directions in order to show comparable results to T11's center-out-and-back task with eight peripheral targets. We further quantified the amount of task-related neural activity in each session by comparing the variance accounted for (VAF) by the top two task-related neural components from the first session. VAF was computed by

$$R^2 = \frac{||\bar{Y}||^2 - ||\bar{Y} - FD\bar{Y}||^2}{||\bar{Y}||^2}$$

Where $\bar{Y}$ is the trial-average neural features across conditions on a subsequent day, and encoder components $F$ and decoder components $D$ are estimated from the first session.

## Statistics and reproducibility

All statistical tests were reported using either Pearson's or Spearman's correlation coefficients for correlation, or Wilcoxon rank sum for across days comparison; $p$-values are reported along with the name of the statistical test. A number that follows the "±" sign is a standard deviation. Blinding and randomization were not relevant for this two-participant study.

## Reporting summary

Further information on research design is available in the Nature Portfolio Reporting Summary linked to this article.

## Data availability

All data required to reproduce the findings in this study are publicly available on Dryad (https://doi.org/10.5061/dryad.n2z34tn5s). The dataset contains intracortical neural signals recorded from both participants along with detailed information about the BCI behavior tasks and performance metrics.

## Code availability

The code for reproducing the figures is made available at https://github.com/ewinapun/MINDFUL.

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

## Acknowledgements

The authors would like to thank participants T11, T5, their families and caregivers, Beth Travers, Dave Rosler, Maryam Masood, Ronnie Gross, Kristi Emerson, Sandrin Kosasih, Beverly Davis, and Kathy Tsou for their contributions to this research. We acknowledge Krishna Shenoy for his inspiration and effort toward creating an environment that spawned this work. Research was supported by NIH-NIDCD (U01DC017844), NIH-NIMH (T32MH115895), NIH-NIDCD (R01DC014034), NIH-NINDS (UH2NS095548), the Office of Research and Development, Rehabilitation R&D Service, Dept of Veterans Affairs (A2295R, A2827R, A3803R, N2864C), the Croucher Foundation, Larry and Pamela Garlick, and Wu Tsai Neurosciences Institute.

## Author contributions

T.K.P., M.T.H., and L.R.H. initiated the study. T.K.P., M.K., and M.T.H. designed and investigated the methodology. T.K.P. conducted the analysis. M.K. performed additional statistical analysis. T.K.P., T.H., G.H.W., A.K., and F.K. were responsible for data collection. A.K. and F.K. were responsible for session scheduling, logistics, and equipment setup/disconnection, and T.K.P., T.H., and G.H.W. implemented code, designed the tasks and

sessions, and curated the datasets. T.K.P. drafted the manuscript, and C.E.V., M.T.H., and L.R.H. provided feedback and revisions. All authors reviewed and edited the final manuscript. J.M.H., J.D.S., and L.R.H. were responsible for funding acquisition and clinical research oversight. The study was co-supervised and guided by M.T.H., and L.R.H.

## Competing interests

The authors declare the following competing interests: The content is solely the responsibility of the authors and does not necessarily represent the official views of NIH or the Department of Veterans Affairs or the United States Government. The MGH Translational Research Center has a clinical research support agreement (CRSA) with Axoft, Neuralink, Neurobionics, Precision Neuro, Synchron, and Reach Neuro, for which L.R.H. provides consultative input. L.R.H. is a co-investigator on an NIH SBIR grant with Paradromics, and is a non-compensated member of the Board of Directors of a nonprofit assistive communication device technology foundation (Speak Your Mind Foundation). Mass General Brigham (MGB) is convening the Implantable Brain-Computer Interface Collaborative Community (iBCI-CC); charitable gift agreements to MGB, including those received to date from Paradromics, Synchron, Precision Neuro, Neuralink, and Blackrock Neurotech, support the iBCI-CC, for which LRH provides effort. G.H.W. is a consultant for Artis Ventures. J.M.H. is a consultant for Neuralink and Paradromics and is a shareholder in Maplight Therapeutics and Enspire DBS. He is also an inventor of intellectual property licensed by Stanford University to Blackrock Neurotech and Neuralink.
