## [Peer Review File · Communications Biology]

Reviewers' comments:

Reviewer #1 (Remarks to the Author):

In their manuscript “Measuring instability in chronic human intracortical neural recordings towards stable, long-term brain-computer interfaces”, Pun and co-authors propose a method to monitor long-term online cursor control in intracortical brain-computer-interfaces (iBCIs).

Cursor control in patients with iBCI can be achieved via online decoding of neuronal signals. Decoders commonly require a calibration phase consisting of target acquisition tasks where the presumed motor intention is available. After the calibration session, many factors, including for example array micro-movements, material degradation, or changes in the neural tuning profiles, can induce a persistent decrease in decoding accuracy. Most solutions require both neuronal signals and true movement intention to be known. Nevertheless, the latter is unavailable in uncued iBCI cursor control.

The authors introduce an alternative approach (MINDFUL) derived by calculating the Kullback-Leibler divergence between two distributions of neural features: the reference distribution for which decoding performance is known to be good (e.g., during the calibration session) and a comparison distribution for which performance is unknown. Compelling evidence that the MINDFUL score strongly correlates with decoding accuracy is provided in the manuscript, thus qualifying MINDFUL as a viable technique for online monitoring of iBCI decoders. Furthermore, the authors explore how selecting different features to calculate the MINDFUL score, derived from electrophysiological recordings or the trained decoder's output velocity vectors, influences the score's correlation to cursor control performance.

This manuscript offers a potential solution to detect uncued cursor control degradation in long-term iBCI use. The method proposed is described in great detail, and enough evidence has been presented to support its validity. Moreover, the MINDFUL score enriches previous work on characterizing neuronal signals' temporal variability even outside of the specific context of this research.

I have no principled objections regarding the study's methods and results. I think the authors have clearly illustrated how their work fits in the relevant application area, addressing some of the challenges of long-term use of chronic intracortical BCIs.

I have two important points, however, that I would like to see addressed in a revised manuscript. Both pertain to the proposed method's interpretability and generalizability.

(1) In the results sections “MINDFUL correlates with decoding performance” and “Correlation to performance increases by combining neural features and decoder outputs” (Figures 1 and 2, respectively), we can observe a reversed effect of testing different features to compute the MINDFUL score correlation to the median angle error (AE) in the two participants. For participant T11, the MINDFUL score derived from the neural features reaches a strong significant correlation to the AE (Pearson = 0.91, Spearman = 0.9), which is not substantially improved by including the velocity features output of the decoder (Pearson = 0.926, Spearman = 0.913). On the other hand, for participant T5, the MINDFUL score derived from the velocity features is strongly significantly correlated to the AE (Pearson = 0.702, Spearman = 0.765) and is not substantially improved by adding the neural features (Pearson = 0.719, Spearman = 0.759).

How methodological differences could explain these results is not further investigated or speculated in these sections or the discussion. I suggest extending the description of the variability of these results among the two participants enrolled in the study.

For example, for participant T5 a linear decoder was trained using spike rates as input during a random target selection task, whereas for participant T11 a non-linear LSTM decoder was trained on spike rates and power during a center-out-and-back task. Such differences support the robustness and generalizability of the proposed method by showing that it yields good results in variable settings, which could be further emphasized.

(2) The selection of the reference distribution for participant T5 has a more significant impact compared to T11. We can see some indication of such discrepancy in Figure 5a, where selecting a reference distribution with high AE (50°-100° or 100°-180°) has a more drastic effect on the correlation between MINDFUL and AE for participant T5. Furthermore, only extracting the reference distribution from both the first and second sessions led to a strong correlation between MINDFUL (computed from neural features only) and AE in T5, as we can observe by comparing Figure 1c (first and second sessions) and Supplementary Figure 6a (first session only).

Although briefly discussed, it would be beneficial for replication purposes to further elaborate how such effects could be traced back to the specific procedures followed for participant T5.

Reviewer #2 (Remarks to the Author):

General comments:

The paper demonstrates the utility of a statistical method for quantifying decoder instabilities, which is an important challenge facing the deployment of brain-computer interfaces. Eventually, BCI's will be used in the community and without supervision by trained researchers. Since decoding performance tends to vary over time, it is necessary to detect if/when decoder performance is degrading. An automated method for doing so would be extremely useful and absolutely necessary in cases where the BCI is used by patients with no other ability to communicate except via the BCI.

The paper is extremely well-written. The data analysis and statistical methods are rigorous and described clearly, and the figures are dense, but well-organized.

This is a minor comment, but I'm wondering if "model drift" is an accurate label for describing progressive degradation in the prediction accuracy of the model in decoding motor intent. Since the "drift" in this context is due to changes in neural activity rather than the model, the authors may consider a different label for this phenomenon. Signal or feature drift may be more appropriate.

The KLD metric appears to correlate nicely with decoder performance, and the discussion section discusses a variety of ways the measure might be implemented. One question, which I don't see mentioned, is what criteria one might use when specifying a threshold for triggering a recalibration.

While not absolutely necessary, it would be helpful to see data on how the KLD measure shifts following recalibration. After all, the KLD is proposed as a method for triggering recalibration or updating of the decoder. It would be good to verify that the KLD drops after recalibration.

Is there a way to qualify the number of electrodes that have "drift"

Specific comments and questions:

Figure 1b shows that the correlation between KLD and directional error (AE) is highly linear, indicating that KLD is a reliable indicator of decoder performance. Indeed, Figure 1c also shows that the KLD measure generally tracks the day on day variations in AE. However, I'm curious to know why the range of KLD values in Figure 1b is only [0, .3] while the range of

KLD values in Figure 1c is [0, ~2.75] (for T11) and [0, ~0.9] for T5.

Does the relationship between KLD and AE vary at all with movement direction? In other words, does KLD provide an unbiased estimate of decoder performance?

Typo on Page 18 - extra space in “intention”

Reviewer #3 (Remarks to the Author):

This study developed a method for assessing model drift or non-stationarities in on-line iBCI control that relies on the underlying neural population activity distribution without having to rely on the performance of BCI control. It relies on computing the Kullback-Leibler divergence (KLD) between the population activity distribution on a given day as compared to a reference population distribution and was validated by comparing the KLD with the angular error between the instantaneous decoded velocity and the inferred intended direction vector (cursor to target position). They found that that the KLD correlated well with the angular error and even better when the neural population activity was augmented with the decoded velocity and its lagged velocity. They then examined changes in directional tuning and found that the KLD computed between sessions negatively correlated with the similarity of the tuning properties of the electrode channels between sessions. Finally, using demixed PCA, they identified two dimensions that separated neural responses across movement directions on the first recording sessions and then projected neural data on these dimensions on successive sessions and found that unexpectedly the neural trajectories became less distinct over successive recording sessions. Moreover, the fraction of variance accounted for in these 2 dimensions decreased with successive sessions and correlated with the KLD.

Overall, this was an interesting and well done study. My major concern, however, is the following:

1. For this approach to be useful for determining when to recalibrate, it should be insensitive to the task being performed. However, it seems to me that the statistics of the neural population would change across very different tasks. If this is true, MINDFUL might erroneously indicate model drift when, in fact, there was no model drift when the subject switches tasks. This concern is partially addressed with T5 given that each session presumably has a different set of random targets. However, what if the task changed more

dramatically. T11 had to perform a center-out task with a click which may be qualitatively different enough from the random target pursuit task that T5 performed. What if you switched from the center-out task with a click to a random target pursuit task in the same subject in the same session when there would be little chance of model drift? This may not be easily done now with the human subjects for practical reasons, but there may be other approaches to address this concern. For example, within one session in the center-out task, one could compute the KLD between trials to one target direction versus trials to another target direction and compare that with KLD between trials to a target and other trials to the same target. My guess is that KLD would be higher for different targets as compared to the same targets despite similar angular errors.

Minor concerns.

1. In Figure 4, how is KLD computed for decoder day 0? Isn't KLD computed between a decoder day and day 0. How is it done within day 0?
2. It is stated that data are collected on 15 consecutive sessions spanning 142 days for T11 and 6 consecutive sessions spanning 28 days for T5. Does this mean that sessions were not performed on many days within these large spans of time?
3. References to Figure 3 panels d-e presented after 3f-h.

Dear reviewers,

We would like to thank the reviewers for their thorough evaluation of our manuscript and for their constructive feedback. R1 highlighted the varying influence of different predictive factors in the MINDFUL metric across participants, possibly linked to methodological differences. To address this, we have included a new paragraph in the manuscript that specifically discusses the interpretability and generalizability of our proposed method. We describe the methodological differences in our datasets from two participants and how they may contribute to the differences observed in KLD values. This new paragraph also helped address Reviewer #2's comment on the observed range of KLD values.

Both R1 and R2 raised an excellent point regarding the impact of reference distribution selection. We have incorporated their observation into the original paragraph related to reference selection for improved clarity. Additionally, a new analysis has been added to the supplemental material to investigate the effect of sub-selecting based on movement direction within the reference distribution.

Lastly, R3 raised a concern about the possibility of MINDFUL mistaking context differences during task switching for model drift. We have included more data from the same sessions of participant T11. This data showcases different 2D cursor tasks being performed using the same decoder. We have added a new results section that highlights the robustness of MINDFUL when using different 2D cursor control tasks as the MINDFUL reference.

Below we address each question or concern separately. These valuable suggestions helped us further strengthen the manuscript. Thank you for devoting your time and attention to our work.

Best regards,
Tsam Kiu Pun

Note: reviewers' comments appear in **black text**. Our replies appear in **blue text**, and revised manuscript text appears indented (with old text shown in **black** and new edits in **red**).

Reviewer #1 (Remarks to the Author):

In their manuscript “Measuring instability in chronic human intracortical neural recordings towards stable, long-term brain-computer interfaces”, Pun and co-authors propose a method to monitor long-term online cursor control in intracortical brain-computer-interfaces (iBCIs).

Cursor control in patients with iBCI can be achieved via online decoding of neuronal signals. Decoders commonly require a calibration phase consisting of target acquisition tasks where the presumed motor intention is available. After the calibration session, many factors, including for example array micro-movements, material degradation, or changes in the neural tuning profiles, can induce a persistent decrease in decoding accuracy. Most solutions require both neuronal signals and true movement intention to be known. Nevertheless, the latter is unavailable in uncued iBCI cursor control.

The authors introduce an alternative approach (MINDFUL) derived by calculating the Kullback-Leibler divergence between two distributions of neural features: the reference distribution for which decoding performance is known to be good (e.g., during the calibration session) and a comparison distribution for which performance is unknown. Compelling evidence that the MINDFUL score strongly correlates with decoding accuracy is provided in the manuscript, thus qualifying MINDFUL as a viable technique for online monitoring of iBCI decoders. Furthermore, the authors explore how selecting different features to calculate the MINDFUL score, derived from electrophysiological recordings or the trained decoder's output velocity vectors, influences the score's correlation to cursor control performance.

This manuscript offers a potential solution to detect uncued cursor control degradation in long-term iBCI use. The method proposed is described in great detail, and enough evidence has been presented to support its validity. Moreover, the MINDFUL score enriches previous work on characterizing neuronal signals' temporal variability even outside of the specific context of this research.

I have no principled objections regarding the study's methods and results. I think the authors have clearly illustrated how their work fits in the relevant application area, addressing some of the challenges of long-term use of chronic intracortical BCIs.

We appreciate that the reviewer believes this manuscript is of importance for clinical translation of iBCIs for long-term use.

I have two important points, however, that I would like to see addressed in a revised manuscript. Both pertain to the proposed method's interpretability and generalizability.

(1) In the results sections “MINDFUL correlates with decoding performance” and “Correlation to performance increases by combining neural features and decoder outputs” (Figures 1 and 2, respectively), we can observe a reversed effect of testing different features to compute the MINDFUL score correlation to the median angle error (AE) in the two participants. For participant T11, the MINDFUL score derived from the neural features reaches a strong significant correlation to the AE (Pearson = 0.91, Spearman = 0.9), which is not substantially improved by including the velocity features output of the decoder (Pearson = 0.926, Spearman = 0.913). On the other hand, for participant T5, the MINDFUL score derived from the velocity features is strongly significantly correlated to the AE (Pearson = 0.702, Spearman = 0.765) and is not substantially improved by adding the neural features (Pearson = 0.719, Spearman = 0.759).

How methodological differences could explain these results is not further investigated or speculated in these sections or the discussion. I suggest extending the description of the variability of these results among the two participants enrolled in the study.

For example, for participant T5 a linear decoder was trained using spike rates as input during a random target selection task, whereas for participant T11 a non-linear LSTM decoder was trained on spike rates and power during a center-out-and-back task. Such differences support the robustness and generalizability of the proposed method by showing that it yields good results in variable settings, which could be further emphasized.

Thank you for your suggestion. It's encouraging that despite a number of methodological differences between the datasets and chosen features, our proposed method still correlates highly to performance. We agree that MINDFUL is somewhat agnostic to these methodological differences, but we hesitate to conclude that MINDFUL is fully generalizable based only on two datasets from two participants where there are a number of variable differences between the two. To make this more clear and to ensure that the current limitations are highlighted, we have added a paragraph on line 446 to highlight the difference between the two datasets in the Discussion section. New text includes:

“While the MINDFUL score based on KLD consistently correlates with performance and changes in neural representation, there are a number of noticeable differences in the results between subjects. First, the relevance of the chosen neural features appears to be different for different participants. For example, for participant T11, the MINDFUL score derived from the neural features is more strongly correlated to the AE than the MINDFUL score derived from decoded velocity features, slightly improving when combining both features; for participant T5, the MINDFUL score derived from the decoded velocity features is more strongly correlated to the AE than neural features alone and is not substantially improved by adding the neural features (Fig 1c and 2). The

features used, choice of decoder and cursor task, and duration of data collection, can all influence the value of KLD, hence correlation to AE.

[...]

Despite a number of methodological differences between the datasets and chosen features, it's encouraging that MINDFUL robustly measured neural instability for both participants.”

(2) The selection of the reference distribution for participant T5 has a more significant impact compared to T11. We can see some indication of such discrepancy in Figure 5a, where selecting a reference distribution with high AE (50°-100° or 100°-180°) has a more drastic effect on the correlation between MINDFUL and AE for participant T5. Furthermore, only extracting the reference distribution from both the first and second sessions led to a strong correlation between MINDFUL (computed from neural features only) and AE in T5, as we can observe by comparing Figure 1c (first and second sessions) and Supplementary Figure 6a (first session only).

Although briefly discussed, it would be beneficial for replication purposes to further elaborate how such effects could be traced back to the specific procedures followed for participant T5.

Thank you for raising this point. Similar to the above consideration about feature inputs , we speculate that the difference can be due to subject (or neuronal selection) variability and the experimental differences in our datasets. We agree that this is a good observation that should be included in the paper. We added this paragraph in the discussion on line 467 where we discuss the importance of choosing a reference:

“Choosing an appropriate reference when calculating KLD influences the reliability of decoder performance predictions using MINDFUL. First, selecting low AE time steps for reference was found to provide a higher correlation between the KLD and AE (**Fig. 5a**) [...]

However, there exists subject variability and ambiguity regarding precisely how much data are needed for reference. For example, for participant T5, using timesteps with high AE (50°-100° or 100°-180°) as reference resulted in a more drastic decrease in the correlation between KLD and AE than T11. Also, using the first two sessions as reference resulted in a slightly higher Pearson correlation than just the first session alone for T5 (**Supplementary Fig. 6**).”

Reviewer #2 (Remarks to the Author):

General comments:

The paper demonstrates the utility of a statistical method for quantifying decoder instabilities, which is an important challenge facing the deployment of brain-computer interfaces. Eventually, BCI's will be used in the community and without supervision by trained researchers. Since decoding performance tends to vary over time, it is necessary to detect if/when decoder performance is degrading. An automated method for doing so would be extremely useful and absolutely necessary in cases where the BCI is used by patients with no other ability to communicate except via the BCI.

The paper is extremely well-written. The data analysis and statistical methods are rigorous and described clearly, and the figures are dense, but well-organized.

We appreciate your recognition of our work.

This is a minor comment, but I'm wondering if "model drift" is an accurate label for describing progressive degradation in the prediction accuracy of the model in decoding motor intent. Since the "drift" in this context is due to changes in neural activity rather than the model, the authors may consider a different label for this phenomenon. Signal or feature drift may be more appropriate.

Thank you for raising this concern. We had a lot of discussion and debate on the term to use. We ultimately use model drift because it is a commonly used term for this exact phenomenon in the machine learning literature. Moreover, other terms such as *feature drift* or *nonstationarity* could be misattributed to statements about biological changes in the brain or other changes in the interaction between the recorded neurons and the recording device, rather than the (purposefully etiology-agnostic) signal-decoder relationship. We added a sentence in the introduction on line 71 to help clarify our point:

"We ascribe such persistent changes to *model drift*, which we define as changes in the relationship between recorded neural signals and motor intention. Nonstationarity, feature shift, dataset shift are terms that have been used synonymously in the literature for this type of phenomenon, but they sometimes refer to any changes in the recorded signals rather than changes in the signal-decoder relationship²³⁻³⁰."

The KLD metric appears to correlate nicely with decoder performance, and the discussion section discusses a variety of ways the measure might be implemented. One question, which I don't see mentioned, is what criteria one might use when specifying a threshold for triggering a recalibration.

Thank you for bringing up this important point. While this paper is focused on the offline results of MINDFUL, towards real-time applications, it is important to set an appropriate and user-specific threshold for triggering online recalibration for individual users. We address this point as part of a newly added paragraph to the discussion on line 526:

“Second, as previously described, the range of KLD varies between participants. It will be crucial to set an appropriate threshold for triggering a recalibration for individual users. One possible strategy to set a user-specific threshold would be to initialize a threshold based on a AE cut-off from previously collected datasets and iteratively fine-tune the threshold sensitivity by incorporating user’s feedback.”

While not absolutely necessary, it would be helpful to see data on how the KLD measure shifts following recalibration. After all, the KLD is proposed as a method for triggering recalibration or updating of the decoder. It would be good to verify that the KLD drops after recalibration.

Thank you for your question. The proposed method of MINDFUL measures the KLD compared to a reference, which we set to the first test block when the decoder was first deployed. As we recalibrate the decoder, this reference for measuring KLD is reset to a new first test block when the new decoder is first deployed. Because both the reference data and the subsequent comparison data for computing KLD have changed after recalibration, direct comparison of these KLD values before and after recalibration wouldn’t necessarily be informative.

Is there a way to qualify the number of electrodes that have “drift”

Thank you for your question. As captured by KLD, drift can be attributed to changes in the joint distribution of features, not necessarily individual features. This makes it challenging to identify which electrodes recorded neural features that, based on the KLD alone, meaningfully changed the joint distribution. We can, however, look at individual features in isolation and describe the degree to which each feature’s tuning changed, which we present in figure 3.

Specific comments and questions:

Figure 1b shows that the correlation between KLD and directional error (AE) is highly linear, indicating that KLD is a reliable indicator of decoder performance. Indeed, Figure 1c also shows that the KLD measure generally tracks the day on day variations in AE. However, I’m curious to know why the range of KLD values in Figure 1b is only [0, .3] while the range of KLD values in Figure 1c is [0, ~2.75] (for T11) and [0, ~0.9] for T5.

Thank you for raising this point. We are aware of this difference and agree that it is an important point to address in the Discussion. We modified the Discussion with a new

paragraph, which is related to a similar question from Review 1 above. New text relating to this question is added starting on line 454:

“[...] Second, there are noticeable differences in the range of KLD between subjects ([0, ~2.75] for T11 and [0, ~0.9] for T5 in **Fig 1c**). It is possible that variability between subjects and between the dataset may affect the range of KLD in MINDFUL. Interestingly, the range of KLD calculated by binning by performance is much smaller than calculated across time ([0, ~0.3] for both participants; **Fig 1b**). In **Fig 1b**, since data was collected across a wider range of time, both for the reference and comparison distributions, any model drift would likely cause these distributions to have larger variance and, hence, smaller KLD. In support of this conjecture, we found that the determinant of the empirical covariance matrix of the reference distribution by binning by performance (**Fig 1b**) is 3.4 times larger than the reference set to the first session (**Fig 1c**) for T11, whilst for T5, these determinants are relatively equal. [...]”

Does the relationship between KLD and AE vary at all with movement direction? In other words, does KLD provide an unbiased estimate of decoder performance?

Thank you for your question. We ran additional analysis that focuses on sub-selecting reference time bins based on movement direction (up/down/left/right), and added them in the supplemental material. We also added the following sentence in the Discussion on line 512.

“There are several additional considerations when applying the MINDFUL score online. First, during personal iBCI use, the movement directions could be less symmetric and more sparsely distributed than the cursor tasks used in this study. When sub-selecting reference time bins based on movement direction (up/down/left/right), KLD became higher in magnitude and generally less correlated to AE (see **Supp. Fig 10, 11**). A longer time window or careful time bin selection for both reference and target distributions may be needed to reduce the directional-related differences.”

Supplemental Figure 10. MINDFUL is robust to reference data: KLD calculated using reference sub-selecting based on direction correlates with AE when compared to distributions not sub-selecting or sub-selecting based on the same direction, but is less correlated to distributions sub-selecting based on the opposite directions.

As in **Fig 1b**, samples of neural features were grouped according to decoder performance (AE), combining all sessions. The reference was neural features at time instances with $AE < 4^\circ$ and conditioned to decoded movement going within $\pm 22.5^\circ$ towards one of the four directions centered at **(a)** 0° (left), **(b)** 90° (up), **(c)** 180° (right) and **(d)** 270° (down) for T11, similarly **(e-h)** for T5. The left polar histogram on each panel indicates the distribution of such sub-selecting references. In each panel, the same conditional references were compared to the other 44 bins with increasing AE intervals, under three conditions: (i) “on all” refers to distributions that include all time bins of AE within each given AE interval. “on up/down/left/right” refers to subsampled time steps within the AE intervals that are also moving towards the specified direction (either (ii) same as reference direction or (iii) opposite direction).

The polar histogram appears to be narrower to the center direction for T11 because of the task design: T11 was performing a center-out-and-back task versus T5 was performing a random target task. Under condition (ii), because the reference and the first target distribution are the same (both $AE < 4^\circ$ going in the same direction), KLD starts at 0. Under the other two conditions, since the first target distribution is different, KLD starts at a non-zero value. When compared to target distribution (i) no sub-selecting or (ii) sub-selecting to the same direction as the reference, KLD still linearly tracks with performance, but for (ii), the range of KLD appears to be bigger for T5, and the pearson correlation is higher than (i) for T5 but not for T11. As for (iii) where the compared distributions are going in the opposite direction, there is relatively much less correlation for both T11 and T5, except in panel **(a)**.

Supplemental Fig 11. Uneven sampling of movement directions in the reference distribution decreases correlation between KLD and AE

(a) The KLD between distributions of X and X_{lag} overlaid onto median AE, for T11 and (b) for T5. “All” in orange indicates when the reference is not sub-selecting based on direction (same as Fig. 2b), and other colors indicate the directions where the reference is sub-selecting on. (c) The KLD between distributions of the combination of derived neural features, decoded directional velocity and its lagged velocity, overlaid onto median AE, for T11 and (d) for T5. “All” also indicates using a reference that was not sub-selecting based on direction (as in Fig. 1c) Colors of the plots are consistent with the upper panels. For easier comparison, for each line, all KLDs are normalized by scaling down by the 99.5th percentile of all its data points and the scaling values are shown in the table on panel (e).

The different choices of reference, sub-selected by direction, still highly correlates with AE, although not as highly as the “all” reference for T11, and “Left” and “Right” correlate slightly more highly than the “all” reference for T5. However, the scaling factor is much higher for some directions, and the KLD baseline is also higher.

Typo on Page 18 - extra space in “intentio n”

Thank you for catching these mistakes and our apologies for not spotting them earlier. We have corrected it in the manuscript.

Reviewer #3 (Remarks to the Author):

This study developed a method for assessing model drift or non-stationarities in on-line iBCI control that relies on the underlying neural population activity distribution without having to rely on the performance of BCI control. It relies on computing the Kullback-Leibler divergence (KLD) between the population activity distribution on a given day as compared to a reference population distribution and was validated by comparing the KLD with the angular error between the instantaneous decoded velocity and the inferred intended direction vector (cursor to target position). They found that the KLD correlated well with the angular error and even better when the neural population activity was augmented with the decoded velocity and its lagged velocity. They then examined changes in directional tuning and found that the KLD computed between sessions negatively correlated with the similarity of the tuning properties of the electrode channels between sessions. Finally, using demixed PCA, they identified two dimensions that separated neural responses across movement directions on the first recording sessions and then projected neural data on these dimensions on successive sessions and found that unexpectedly the neural trajectories became less distinct over successive recording sessions. Moreover, the fraction of variance accounted for in these 2 dimensions decreased with successive sessions and correlated with the KLD.

Overall, this was an interesting and well done study. My major concern, however, is the following:

1. For this approach to be useful for determining when to recalibrate, it should be insensitive to the task being performed. However, it seems to me that the statistics of the neural population would change across very different tasks. If this is true, MINDFUL might erroneously indicate model drift when, in fact, there was no model drift when the subject switches tasks. This concern is partially addressed with T5 given that each session presumably has a different set of random targets. However, what if the task changed more dramatically. T11 had to perform a center-out task with a click which may be qualitatively different enough from the random target pursuit task that T5 performed. What if you switched from the center-out task with a click to a random target pursuit task in the same subject in the same session when there would be little chance of model drift? This may not be easily done now with the human subjects for practical reasons, but there may be other approaches to address this concern. For example, within one session in the center-out task, one could compute the KLD between trials to one target

direction versus trials to another target direction and compare that with KLD between trials to a target and other trials to the same target. My guess is that KLD would be higher for different targets as compared to the same targets despite similar angular errors.

Thank you for raising this issue. We recognize that this is an important concern towards practical iBCI application and should be addressed in the paper. Regarding the comparison to different target directions, please see the response to a similar question from Reviewer 2 above and the corresponding results in Supplementary Figure 10 and 11. More importantly, we ran additional experiments to compare the effect of using reference data from a different continuous 2d cursor task (random target task, and personal iBCI use) on the test data task. We apologize that we can only include additional T11 data in this experiment as only T11 performed additional blocks of various tasks on day 0 and day 7, but not for T5. We also are unavailable to collect additional data from T5 due to his recent passing (unrelated to the participation in the clinical trial). Descriptions are as follows.

We added a new section in the **Results** starting on line 357, along with a new figure:

MINDFUL is robust to the reference task. We repeated the analysis of tracking the correlation of MINDFUL to performance (see **Fig 2c**) for T11 except we used reference data collected during different tasks. The comparison data collected during center-out-and-back tasks remains the same as in fig 2c. Since the relationship between neural activity and movement can be context and task dependent (Downey et al. 2017; Gallego et al. 2018), it is unclear to what degree the reference and comparison tasks must be matched for MINDFUL to correlate well to performance. MINDFUL is likely to be most useful in practice if it is robust across tasks and contexts. Collecting reference data from a different task was not part of our original experimental design, but for T11 the appropriate data was collected for other purposes. In addition to the center-out-and-back task, T11 used the same fixed decoder for random target tasks (day 7), analogous to T5's task described above, and during personal iBCI use (i.e. browsing the web; day 0; See **methods**). Despite different cursor tasks being used as the reference from the target distributions, MINDFUL still correlates highly to performance, even without the help of subsampling based on AE (see **Fig 6b-c**). This is also true when all three types of tasks were combined together for estimating the reference distribution (mixed tasks, see **Fig 6d**). The range of KLD is slightly higher for the random target task, and lower for personal use and mixed tasks. This suggests that MINDFUL might be robust to cursor task changes for reference.

Figure 6. MINDFUL reference can be applied across multiple cursor control tasks. Instead of using features from day 0 where participant T11 was performing the same tasks as subsequent days (center-out-and-back), different 2D cursor tasks were selected as reference. In these tasks, T11 used the same LSTM fixed decoder. The same PCA procedure as described in the methods was applied. No subsampling on angle error was performed on the reference as it might not be explicitly available during personal use. Features used include all neural features and decoded kinematics plus its lagged version.

(a) Reference was set to a block of T11 performing a random target task (5 mins) on day 7, using the same fixed decoder. Task setup is very similar to T5’s data except with varying target sizes. T11 did not perform this task on day 0.

(b) Reference was set to when T11 was using the iBCI for personal desktop use, such as browsing the internet, on day 0. Around 10 mins of active cursor control period was included.

(c) Reference was set to center-out task on day 0, same as in figure 2c except AE was not sub-selected to be less than 4°.

(d) Reference was set to a combination of the above mentioned data (concatenating random target task, personal use, and center-out-and-back task).

In Discussion on line 517:

“There are additional several considerations of applying the MINDFUL score online. First, during personal iBCI use, the movement directions could be less symmetric and more sparsely distributed than the cursor tasks used in this study. Context shifts can be an issue in BCI decoder design as they can affect decoder performance. When sub-selecting reference time bins based on uni-directional movements (up/down/left/right), KLD became much higher in magnitude and generally less correlated to AE (see **Supp. Fig 10, 11**). A longer time window or careful time bin selection for both reference and target distributions may be needed to reduce the directional-related differences. **In addition, we believe that the MINDFUL approach may be useful for detecting changes in the relationship between the signal and the decoder(s), even when the multiple disparate tasks and contexts are incorporated into the reference set. For T11, MINDFUL was found to be robust to using neural reference**

data collected during different cursor tasks, including periods of personal iBCI use for which we had no control over the balance of intended directions or angle error. We did not have the data to investigate this in T5. Future work will investigate this robustness in additional participants and more varied changes in tasks and contexts, all of which will be important for practical iBCI use."

In Method on line 683:

MINDFUL robustness to reference data across tasks. In **Fig 6**, we repeated the process of calculating MINDFUL as in **Fig 2c** except using different task data for the reference distribution. Additionally, no subsampling based on AE was applied for the reference as the ground truth performance is not available during personal use (no target was cued and therefore performance metrics are not readily available). Approximately 8 minutes of personal use and 10 minutes of random target tasks were included for the reference. The target distributions were still estimated from a 60-second sliding window during center-out-and-back tasks on subsequent sessions. Thus, median AEs calculated over time bins of these target distributions remain the same as **Fig 1c** and **2c** in this analysis.

Minor concerns.

1. In Figure 4, how is KLD computed for decoder day 0? Isn't KLD computed between a decoder day and day 0. How is it done within day 0?

Thank you for your question. The mean KLD from Fig 4 is computed the same way as described in the results on page 8. We clarified this point with the new text below on line 260:

"To compare with the tuning map correlation between sessions, we obtain a mean KLD between each pair of sessions. Instead of fixing a reference distribution, pairwise KLDs of neural features between sessions were calculated using a sliding window of 60 seconds updating every 10 seconds (see **Methods**). The KLDs from the same session were averaged to get a mean of the neural distribution difference between pairs of sessions."

In **Methods**, we include the following detailed explanation:

"Mean KLD between sessions. To estimate the average neural distribution difference between pairs of sessions, a mean KLD between distributions on two given sessions was calculated (see Fig. 3e, 3j). The 9-dimensional derived neural features are the same as Fig. 2c, 5, namely, the top $M=5$ PCs, X , and X_{lag} . The PCA-reference data is the same as Fig. 1c, 2, and 5, namely, time bins from the first session (T1) or first two sessions (T5) that have $AE < 4^\circ$. The reference and comparison data are each

60-second intervals updated every 10 seconds. If there are M such intervals in session i and N in session j , then the mean KLD between the sessions is the average KLD of all (MN) pairs of intervals from sessions i and j . Outlier trials were excluded from the intervals. For visualizing the complete pairwise comparison matrix containing the KLD of all pairs of intervals from all sessions, see Supplementary Fig. 9. “

We added the following sentence in the results section on line 302 that describe the mean KLD in figure 4 to improve clarity:

“Mean KLD between day 0 and other sessions is calculated the same way as the mean KLD in the previous section (averaged KLDs using a sliding window of 60 seconds updating every 10 seconds; see **Methods**).”

2. It is stated that data are collected on 15 consecutive sessions spanning 142 days for T11 and 6 consecutive sessions spanning 28 days for T5. Does this mean that sessions were not performed on many days within these large spans of time?

Thank you for raising this question. We apologize for the lack of clarity; we shouldn't have used the word “consecutive”. Typically, a session is conducted within a day, we run one to three sessions per week with each participant, and sessions on any given week may have different research focus where the participants perform different tasks. For this study, we present data collected from 15 sessions, which refers to data collected on 15 separate days out of the 142 days for T11. Likewise for T5.

We've modified in the results on line 113 with:

“Data were collected from 15 research sessions, **each from a separate day over a 142 day period**, of T11 performing a center-out-and-back task using a fixed RNN (recurrent neural network) decoder, as previously described.”

3. References to Figure 3 panels d-e presented after 3f-h.

Thank you for raising this point. We have reordered the subfigures in figure 3.

REVIEWERS' COMMENTS:

Reviewer #1 (Remarks to the Author):

All my comments have been addressed. I would like to congratulate the authors on a very interesting paper.

Reviewer #2 (Remarks to the Author):

Thank you for addressing my comments with the rigor and clarity of your original submission.

Reviewer #3 (Remarks to the Author):

The authors have addressed my concerns. However, there appears to be a mistake in the titles of the figure panels of Figure 6. They don't match the descriptions of in the figure legend. What do the titles refer to. I assume they refer to the task used for the reference distribution, but, if this is true, this doesn't match the descriptions in the figure legend.